# Electronic transport in planar atomic-scale structures measured by two-probe scanning tunneling spectroscopy

Marek Kolmer[1,2], Pedro Brandimarte [3], Jakub Lis[1], Rafal Zuzak [1], Szymon Godlewski [1], Hiroyo Kawai[4], Aran Garcia-Lekue[3,5], Nicolas Lorente[6], Thomas Frederiksen [3,5], Christian Joachim[7], Daniel Sanchez-Portal[6] & Marek Szymonski[1]

Miniaturization of electronic circuits into the single-atom level requires novel approaches to characterize transport properties. Due to its unrivaled precision, scanning probe microscopy is regarded as the method of choice for local characterization of atoms and single molecules supported on surfaces. Here we investigate electronic transport along the anisotropic germanium (001) surface with the use of two-probe scanning tunneling spectroscopy and first-principles transport calculations. We introduce a method for the determination of the transconductance in our two-probe experimental setup and demonstrate how it captures energy-resolved information about electronic transport through the unoccupied surface states. The sequential opening of two transport channels within the quasi-one-dimensional Ge dimer rows in the surface gives rise to two distinct resonances in the transconductance spectroscopic signal, consistent with phase-coherence lengths of up to 50 nm and anisotropic electron propagation. Our work paves the way for the electronic transport characterization of quantum circuits engineered on surfaces.

[1] Centre for Nanometer-Scale Science and Advanced Materials, NANOSAM, Faculty of Physics, Astronomy and Applied Computer Science, Jagiellonian University, Łojasiewicza 11, PL 30-348 Kraków, Poland. [2] Center for Nanophase Materials Sciences, Oak Ridge National Laboratory, Oak Ridge, Tennessee 37831, USA. [3] Donostia International Physics Center, DIPC, Paseo Manuel de Lardizabal 4, E-20018 Donostia-San Sebastián, Spain. [4] Institute of Materials Research and Engineering, 2 Fusionopolis Way, Innovis, #08-03, Singapore 138634, Singapore. [5] IKERBASQUE, Basque Foundation for Science, E-48013 Bilbao, Spain. [6] Center for Materials Physics CSIC-UPV/EHU, Paseo Manuel de Lardizabal 5, E-20018 Donostia-San Sebastián, Spain. [7] Nanoscience Group & MANA Satellite, CEMES/CNRS, 29 rue Marvig, BP 94347, 31055 Toulouse, France. These authors contributed equally: Marek Kolmer, Pedro Brandimarte. Correspondence and requests for materials should be addressed to M.K. (email: kolmerma@ornl.gov) or to D.S-P. (email: daniel.sanchez@ehu.eus)

The invention of the scanning tunneling microscope (STM) by Binnig et al.[1,2] opened a new era in surface science. It is now a standard microscopy technique for real-space imaging of the electronic structure of conducting surfaces with picometer resolution[2–4]. Single-probe STM is also a spectroscopic tool, able to locally probe electronic surface states as a function of the bias voltage in the scanning tunneling spectroscopy (STS) mode[5,6]. Furthermore, the precision reached in approaching the STM tip apex toward the surface permits for a controlled electronic contact with a single surface atom or molecule[7,8]. Thus, such vertical contacts formed by STM can be used to study electronic transport through adsorbates with atomic-scale lateral resolution[9–14].

Direct determination of the electronic transport properties of a planar atomic-scale wire or circuit lies beyond the single-probe approach. Such characterization requires fabricating metal contacts with high precision[15–18], which is usually a challenge. An attractive alternative is the use of multi-probe STM[19–22]. This latter method offers high control on the position and geometry of the contacts between the probes and the nanoscale system. However, downscaling of multi-probe instruments toward the atomic level, i.e., where all STM tip apex positions are controlled at the atomic scale, meets many technical obstacles. Although two-probe STM (2P-STM) experiments have already been proposed[23–26], practical implementations of those propositions were not reported so far. Recent technical advances, however, offer a new generation of multi-probe instruments with STM tips operating simultaneously on the same surface and with a stability comparable to the best cryogenic single-probe STMs[27]. In fact, only recently 2P-STM experiments have reached the required atomic precision in contacting structures on a surface[28]. That technical result made atomic-scale 2P-STM experiments feasible; but, to date, no experimental protocols for extracting transport properties of atomic structures from such experiments have been reported.

In this work, we directly observe quasi-one-dimensional (1D) electronic transport channels provided by the unoccupied surface states running along the dimer rows on the Ge(001) surface. Understanding the transport properties of this surface is important, as it provides an excellent platform for fast and reliable fabrication of atomic-scale circuits. This can be achieved, e.g., by STM-induced selective hydrogen desorption from the hydrogen passivated Ge(001):H surface[28–32]. Our experiments were made using a specific 2P-STM/STS approach allowed by an atomically precise STM probe positioning protocol with relative probe-to-probe separation distances down to 30 nm. Our 2P-STS identification of the transport channels is corroborated further by: single-probe $dI/dV$ STS characterization of the electronic states of a Ge dimer row next to a monoatomic Ge(001) step edge; first-principles calculations using density functional theory (DFT), and multi-terminal transport calculations performed using non-equilibrium Green's functions (NEGF). We thus show that planar, phase-coherent electronic circuits can be achieved on reconstructed Ge(001). Besides this exploration of prototypical atomic circuits, our measurement protocol provides a general tool to explore in-plane electronic transport applicable to high interest research fields including for example engineered two-dimensional (2D) systems[22,33–38] or materials with topological electronic states[39–42].

## Results

**Two-probe STS experiment.** The Ge(001) surface consists of buckled Ge dimers forming well-separated parallel rows. The existence of surface dangling bonds introduces additional unoccupied states within the band-gap of the bulk Ge electronic structure[43–46]. The dispersive surface conduction bands of interest are formed by the interaction between the $\pi^*$ orbitals of Ge dimers along the rows and lie mostly inside the bulk electronic gap[29] (see Fig. 1a and detailed analysis in Supplementary Note 8). Importantly, weak interactions between adjacent rows result in strong anisotropy of this band structure. Consequently, the reconstructed dimer rows on the bare Ge(001) surface form a series of parallel quasi-1D wires[44–46].

To study the conduction channels introduced by a single Ge dimer row, we follow the experimental protocol presented in Fig. 1b, c. This includes the focused ion beam preparation of the STM probes to be able to control their approach down to an inter-apex separation distance limit of about 30 nm[28]. Marked as tip1 in Fig. 1, the first STM probe is kept in a tunneling regime with a low-bias junction resistance larger than 100 GΩ (see Supplementary Note 2 for details in probe-to-surface contact resistance determination). This junction has the role of a source probe, injecting hot carriers[47] into the Ge electronic states of the selected single row. With about 5 pm precision, the tuning of the tip1-Ge surface distance controls the corresponding tunneling junction resistance and therefore the current intensity through this junction. Marked as tip2 in Fig. 1, the second STM probe is the drain probe kept at a low-bias tunneling junction resistance in the range of tens to hundreds of MΩ. The corresponding junction is maintained in this low-resistance regime with the tip2 positioned over the very same surface row as tip1. In this setup, the Ge(001) sample and the drain STM probe (tip2) are grounded during the whole experiment and the bias voltage is applied only to the source probe (tip1). As mentioned above, a similar two-probe experimental scheme was proposed 20 years ago[23,24], but never realized in practice to the best of our knowledge. During our 2P-STS experiment both STM feedback loops are off and the corresponding tunneling currents are measured using the two available STM $I–V$ converters.

We applied this experimental procedure on the atomically perfect Ge(001)-c(4 × 2) surface area presented in Fig. 2a. Both tips were approached over the very same Ge dimer row at the locations marked by the two circles. Tip1 (blue circle) was kept in a tunneling condition ($I = 10$ pA, $V_{sample} = -0.5$ V). Starting from comparable feedback conditions (20 pA, −0.5 V), tip2 (white circle) was approached down to the surface by 4 Å, resulting in a final ~ 50 MΩ low-bias junction resistance. Fig. 2b shows the simultaneously measured $I_1$ and $I_2$ currents as a function of the tip1 voltage ($V_1$). It is noteworthy that bias voltages in Fig. 2 are defined in reference to the grounded sample, i.e., unoccupied states are probed with $-V_1$ being positive. As the sample is grounded, the $I_1(V_1)$ characteristics (red in Fig. 2b) exhibits a shape comparable to that usually recorded by a single-probe STM on a bare Ge(001) surface. Importantly, we also detect a non-zero $I_2$ current (black in Fig. 2b) for positive values of $-V_1$, in the order of 10% of $I_1$. We assign both $I_1$ and $I_2$ currents as being positive when the current flows from the tip to the sample. Therefore, as clearly noticed in Fig. 2b, when the current is injected from the source tip1 to the sample, a fraction of the current is recorded by the drain tip2 with a negative sign (from sample to tip).

More details on those $I(V)$ curves are found by recording at the same time the corresponding differential $dI/dV$ spectra as seen in Fig. 2c. As expected, the red $-dI_1/dV_1$ spectrum resembles single-probe STS spectra available in the literature for the bare Ge(001)-c(4 × 2) surface with two clear resonances at 0.35 V and 1.1 V[45,48]. As will be clarified later, our transport simulations indicate that they can be assigned to the two surface conduction band edges CBE and CBE + 2 as indicated in Fig. 1a. More importantly, the $dI_2/dV_1$ transconductance spectrum also shows pronounced resonances in the energy range of the Ge dimer

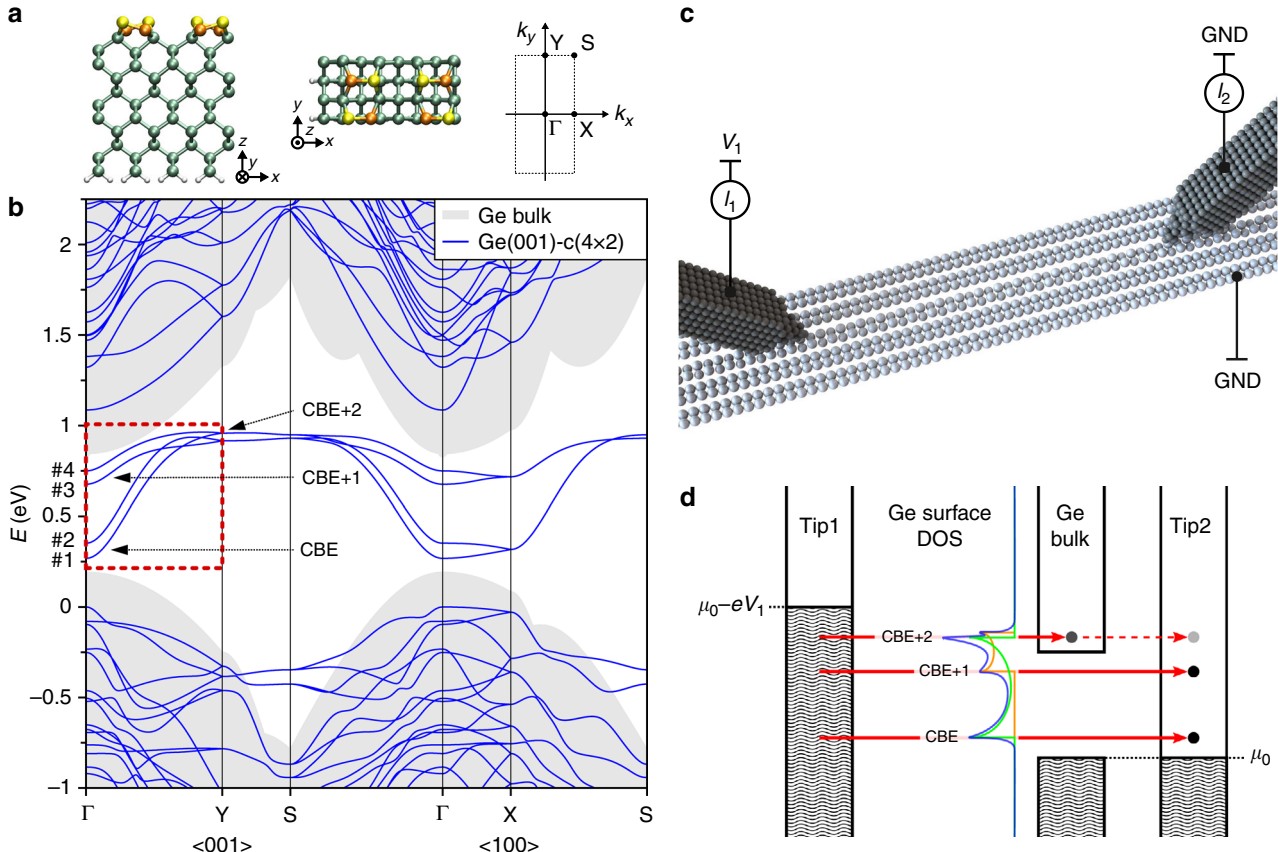

**Fig. 1** Electronic structure of Ge(001)-c(4 × 2) surface and two-probe experimental setup. **a** Side and top views of the used unit cell, twice the size of the primitive cell, and the corresponding Brillouin zone (see Supplementary Fig. 16 for details). Yellow and orange colored Ge atoms highlight the buckled wire. **b** Calculated band structure of a 12-layer Ge(001)-c(4 × 2) slab. The red box highlights the dispersive surface-state bands #1–4. Positions of the conduction band edges CBE (bottom of bands #1 and #2), CBE+1 (bottom of bands #3 and #4) and CBE+2 (top of bands #1–4) correspond to the resonances in the projected density of states (see Supplementary Fig. 17 for details). The bulk Ge band structure is shown in shaded gray (details of the band alignment of slab and bulk are presented in Supplementary Note 10). Notice that the CBE+2 position overlaps with the onset of the bulk conduction band. **c** Model presentation of the experimental setup. Both tips are kept in the tunneling regime above a grounded sample. Bias voltage is applied to tip1, whereas tip2 is virtually grounded through its preamplifier. Corresponding currents are registered on both tips. **d** Two-probe measurement scheme for the transconductance $dI_2/dV_1$ signal, which probes the energy positions of ballistic transport channels mediated by the surface states. The experimental design resembles the ballistic-electron emission microscopy concept[64–66] with tip2 acting as a collector. The schematic surface density of states (blue) shows the three discussed resonances, associated with the edges of the two quasi-1D surface bands (whose density of states are represented in green and orange). Notice that the Fermi energy of Ge(001) is known to be pinned at the top of the Ge bulk valence band[43,44,62,63]. Thus, in our scheme the chemical potentials of tip2 and the surface are aligned

electronic states. At 0.35 V, this first $dI_2/dV_1$ resonance corresponds exactly to the CBE observed also in the vertical $dI_1/dV_1$ recording. Interestingly, for a bias voltage exceeding 0.6 V, the 2P-STS transconductance spectrum is significantly different from the standard vertical single-probe STS spectra. For example, a new $dI_2/dV_1$ resonance appears at 0.7 V, i.e., in the energy range of CBE + 1 (Fig. 1a), which only appears as an elbow in the single-probe $-dI_1/dV_1$. Finally, the CBE + 2 resonance at 1.1 V observed in $-dI_1/dV_1$ is only barely captured in the $dI_2/dV_1$ spectrum. Additional 2P-STS data are presented in Supplementary Fig. 2–4. They include data registered with different pairs of STM probes confirming that general transconductance signal trends are not affected by specific electronic states of the tips.

**Electronic structure of the system**. A distinct signature of coherent propagation is the formation of oscillations in the local density of states (LDOS) close to defects as observed in single-probe STM experiments[49–53]. These oscillations result from interferences between the incoming and elastically scattered

carriers. As our transconductance results are consistent with the picture that electrons propagate elastically along the rows, in order to gain further understanding we performed single-probe STM experiment on a clean Ge(001)-c(4 × 2) surface area near a single monoatomic step-edge (Fig. 3a, with structural details in Supplementary Note 6). This configuration allows detecting characteristic standing wave patterns observed on STS $dI/dV$ maps for positive sample bias voltages (unoccupied states)[44,45,54]. These patterns reflect the energy-dependent electron wavelength and decay slowly, while scanning far away from the step-edge (Fig. 3b). Importantly, the interference patterns are formed only if the coherence of the corresponding electronic waves is preserved. Moreover, due to the electronic decoupling of the corresponding $\pi^*$ states dispersing along the Ge dimer rows from the bulk electronic states (Fig. 1a), the resulting patterns are observed at distances up to 25 nm away from the step edge (Fig. 3c) and for relatively high energies as compared with metal substrates[49–51]. This also suggests that the effective coherence length for the electronic waves along the Ge dimer rows is around twice the distance where we observe the LDOS modulation patterns, i.e., up

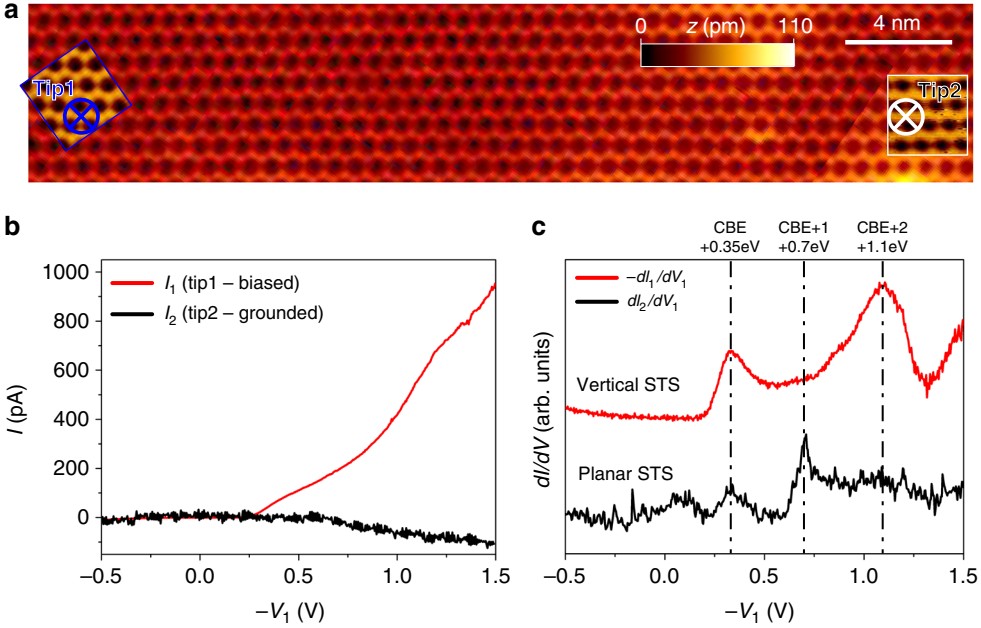

**Fig. 2** Two-probe scanning tunneling spectroscopy characterization of Ge dimer row. **a** Filled-state scanning tunneling microscopy (STM) images of the c(4 × 2) reconstructed Ge(001) surface (I = 20 pA, V = − 0.5 V) obtained before the two-probe scanning tunneling spectroscopy (2P-STS) experiment. Insets show two atomically resolved STM images obtained simultaneously by both probes. STM probe positions for 2P-STS about 30 nm apart over the very same Ge dimer row are marked by blue (tip1) and white (tip2) circles. **b** Current vs. tip1 voltage curves obtained simultaneously on source (tip1, red) and drain (tip2, black) probes. It is noteworthy that during the data acquisition, the sample was grounded and the tunneling contact resistance of tip2-sample junction was established to be ~ 50 MΩ. **c** Corresponding vertical − $dI_1/dV_1$ (see text for sign convention) and planar $dI_2/dV_1$ 2P-STS signals as a function of tip1 voltage obtained with the application of a protocol with two lock-in amplifiers. The resonances observed in the $dI/dV$ characteristics at energies 0.35 eV, 0.7 eV, and 1.1 eV are ascribed to the CBE, CBE + 1, and CBE + 2 resonances, respectively. The small peak in $dI_2/dV_1$ spectrum located around 0.1 eV has unknown origin and it is also registered by single-probe STS experiments performed on Ge(001)-c(4 × 2) surface[48,62]

to 50 nm for the lower energies. Our observations are in remarkable agreement with our theoretical simulations using a two-terminal model of the step-edge with dimensions comparable to the experiment, where the calculated eigenchannels exhibit similar interference patterns localized at the surface (Supplementary Note 13).

In order to enhance the energy resolution of the single-probe experimental data, we collected a series of $dI/dV$ spectra at the positions marked by the white squares in Fig. 3a. The resulting data presented in Fig. 3d demonstrates the reflection of the corresponding electronic states at the step edge. Right side of Fig. 3e presents the 1D Fourier transforms (FTs) of these STS $dI/dV$ spectra (FT-STS) reproducing in detail the unoccupied band structure of a Ge dimer row. Starting at about 0.3 eV, the surface band corresponding to CBE presents two higher intensities corresponding to the minimum and maximum of the band dispersion perpendicular to the Ge dimer wires, noted as #1 and #2, respectively in Fig. 1a. The energy positions and dispersions are in good agreement with the band structure calculations as shown in Fig. 3e where we present the computed k-resolved density of states (DOS, equation 3 in Supplementary Note 9), which confirms that a higher DOS is expected at the energy onsets of the surface band (when plotted as a function of $k_y$ along the Γ−Y direction). The dispersion of these bands agrees also with previously reported single-probe STS studies[44,45,54] and with a very recent angle-resolved two-photon photoelectron spectroscopy experiment[43]. In addition, the FT-STS shows increased contrast at an energy of about 0.65 eV (CBE + 1), where the second pair of dispersive bands #3 and #4 should be present according to our calculated band structure (Fig. 1a). To complete the analysis related to the CBE and CBE + 1 bands, we obtained the dispersion of the bands from fitting the $dI/dV$ cross-

sections in Fig. 3d (for details, see Supplementary Note 6). This procedure clearly captures the two dispersive surface bands related to CBE and CBE + 1. At 0.9 eV the oscillatory pattern becomes very weak and difficult to identify. Besides the complications associated with multiband contributions, this is a clear signature of a reduced coherence length at those high energies. Fig. 1a indicates that those high-energy states are resonant with the conduction band of bulk Ge (see also discussion in Supplementary Note 10). Thus, electrons injected at the energies of the CBE + 2 resonance will be efficiently scattered into bulk during propagation along the wire and reflection at the step edge, explaining the disappearance of the interference pattern.

Single-probe STS measurements described above confirm the presence of dispersive bands, which favor the transport along the Ge dimer rows. These channels correspond to the CBE and CBE + 1 surface band edges identified in Fig. 1a and are recorded in the planar 2P-STS $dI_2/dV_1$ spectra. At low temperature, they provide coherent electronic transport at least up to about 50 nm in length. This long coherent propagation is also due to the low value of the corresponding quasiparticle effective masses, estimated from parabolic fits to data points shown in Fig. 3e to be ~ 0.18 $m_e$ for bands #1 and #2 (CBE) and ~ 0.35 $m_e$ for bands #3, #4 (CBE+1), $m_e$ being the free electron mass (this is also in good agreement with theory, for details see Supplementary Note 6).

**Transport calculations**. In order to verify our interpretation of coherent, planar transport through the Ge surface states, we compare the experimental 2P-STS results with first-principles transport calculations. In our self-consistent multi-terminal treatment, we considered a model system composed of a twelve-

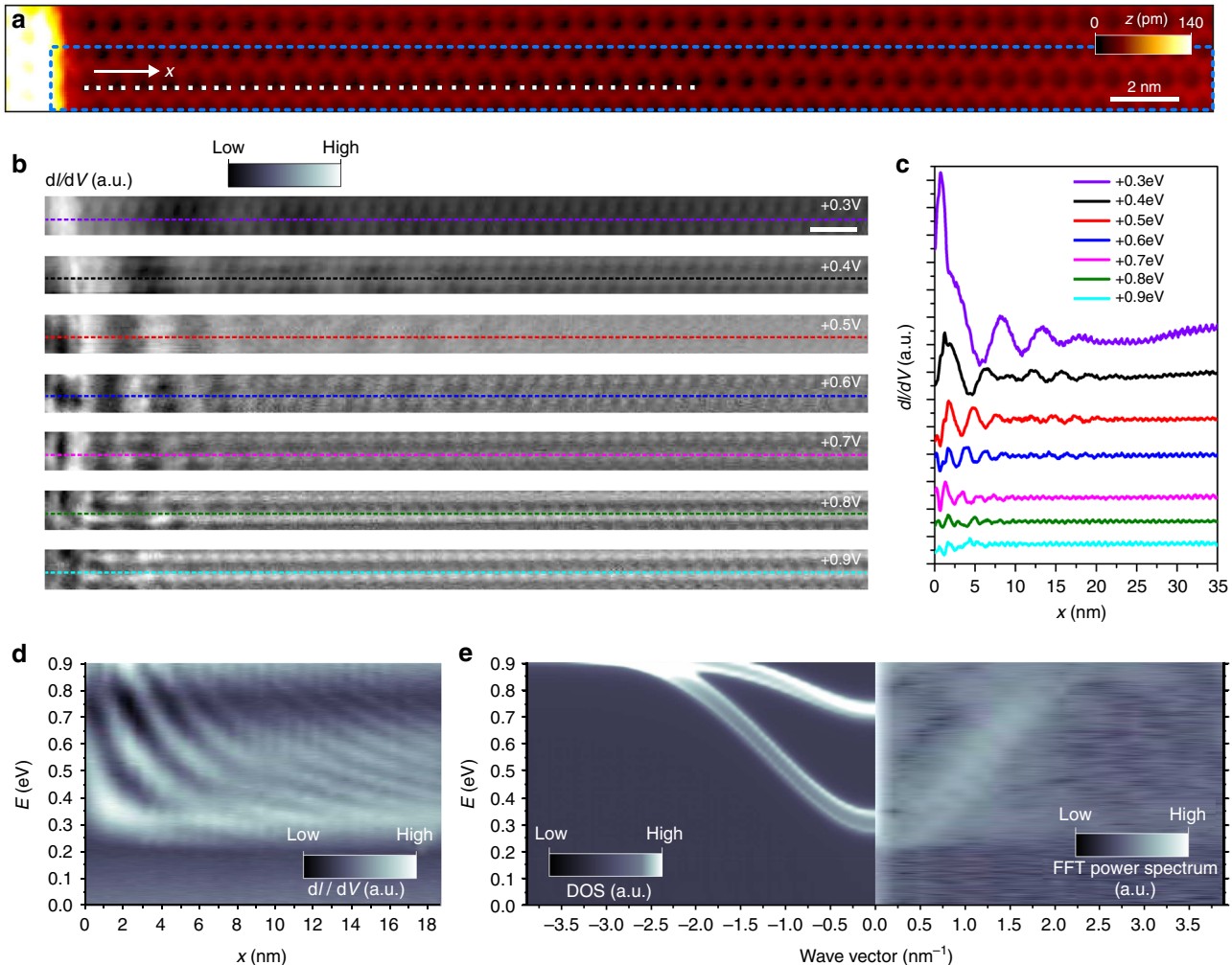

**Fig. 3** Reflection of the quasiparticle wave functions at the step edge. **a** Filled-state scanning tunneling microscopy (STM) image ($I = 100$ pA, $V = -0.5$ V) presenting an atomically perfect surface area near the step edge at Ge(001)-c(4 × 2). **b** Series of empty-state constant-current $dI/dV$ maps (100 pA) obtained on the atomically perfect Ge(001)-c(4 × 2) surface area marked in **a** by the blue box. White bar is 2 nm. **c** Cross-sections of $dI/dV$ maps obtained along dashed lines presented in **b**. Interference patterns related to reflection of electronic waves on the step-edge potential are clearly seen. **d** Single-point scanning tunneling spectroscopy $dI/dV$ data obtained in positions marked by white points in **a**. Brighter contrast represents higher intensity of $dI/dV$ signal. **e** Left: computed density of states of a twelve-layer Ge(001)-c(4 × 2) slab (broadened by $\eta = 0.015$ eV) as a function of energy and $k_y$ (i.e., integrating the contribution for all $k_x$ at each point, see equation 3 in the Supplementary Note 9). Right: one-dimensional Fourier transform of the data presented in **d** reproducing the unoccupied band structure of the Ge dimer wire. Brighter colors correspond to higher intensities

layer Ge(001)-c(4 × 2) slab contacted by Au STM tips oriented along the (100) direction as illustrated in Fig. 4a. Two semi-infinite Ge electrodes were connected at each slab termination (here denoted left and right leads) and the Au tips were positioned perpendicularly to the slab with both tip apexes located atop individual Ge dimers belonging to the same row.

The main results of our transport calculations are summarized in Fig. 4. Further details on these calculations can be found in Supplementary Notes 5, 7 and 10. We first discuss results obtained for a single tip addressing the Ge slab. Fig. 4b shows how the transport along the surface, i.e., the left-to-right transmission function $T_{LR}(E)$, reflects its peculiar band structure and gets disrupted by the presence of the probing tip. For the bare surface (dashed green line), $T_{LR}(E)$ reflects the band structure in Fig. 1a and presents one transmitting channel per Ge dimer row in the range 0.3–0.7 eV and two transmitting channels in the range 0.7–0.9 eV (notice that our supercell contains four dimer rows). When the tip is kept at tunneling distances ($D \geq 4.5$ Å), this result is only weakly modified. It is necessary to approach considerably the tip to the sample ($D \leq 3.5$ Å), in order to find a

significant reduction of $T_{LR}(E)$. The analysis of the transmission probability decomposed in eigenchannels shows that only the Ge dimer row immediately below the tip is significantly affected by its presence. This shows that it is possible to address independently different dimer rows in the surface, as will become clear below.

Besides backscattering of incoming electrons from the Ge leads at the STM tip, the reduction in $T_{LR}(E)$ reflects the opening of the surface-to-tip transmission $T_{st}(E)$. Fig. 4c shows $T_{st}(E)$ for $D = 4.5$ Å, defined as the sum of the transmission probabilities from each of the Ge electrodes into the metallic tip. In this case, the Ge lateral electrodes of the model are playing the same role as the grounded sample bulk supporting the Ge(001) surface. As expected, for tunneling conditions ($D \geq 4.5$ Å), $T_{st}(E)$ resembles the surface PDOS (Fig. 1c and Supplementary Fig.17). $T_{st}(E)$ presents two clear peaks at around 0.3 and 0.9 eV that we assign to the observed CBE and CBE + 2 resonances in the experimental $-dI_1/dV_1$. As mentioned above, the CBE + 2 is resonant with bulk states, which probably has an important contribution to the $-dI_1/dV_1$ spectra at the corresponding

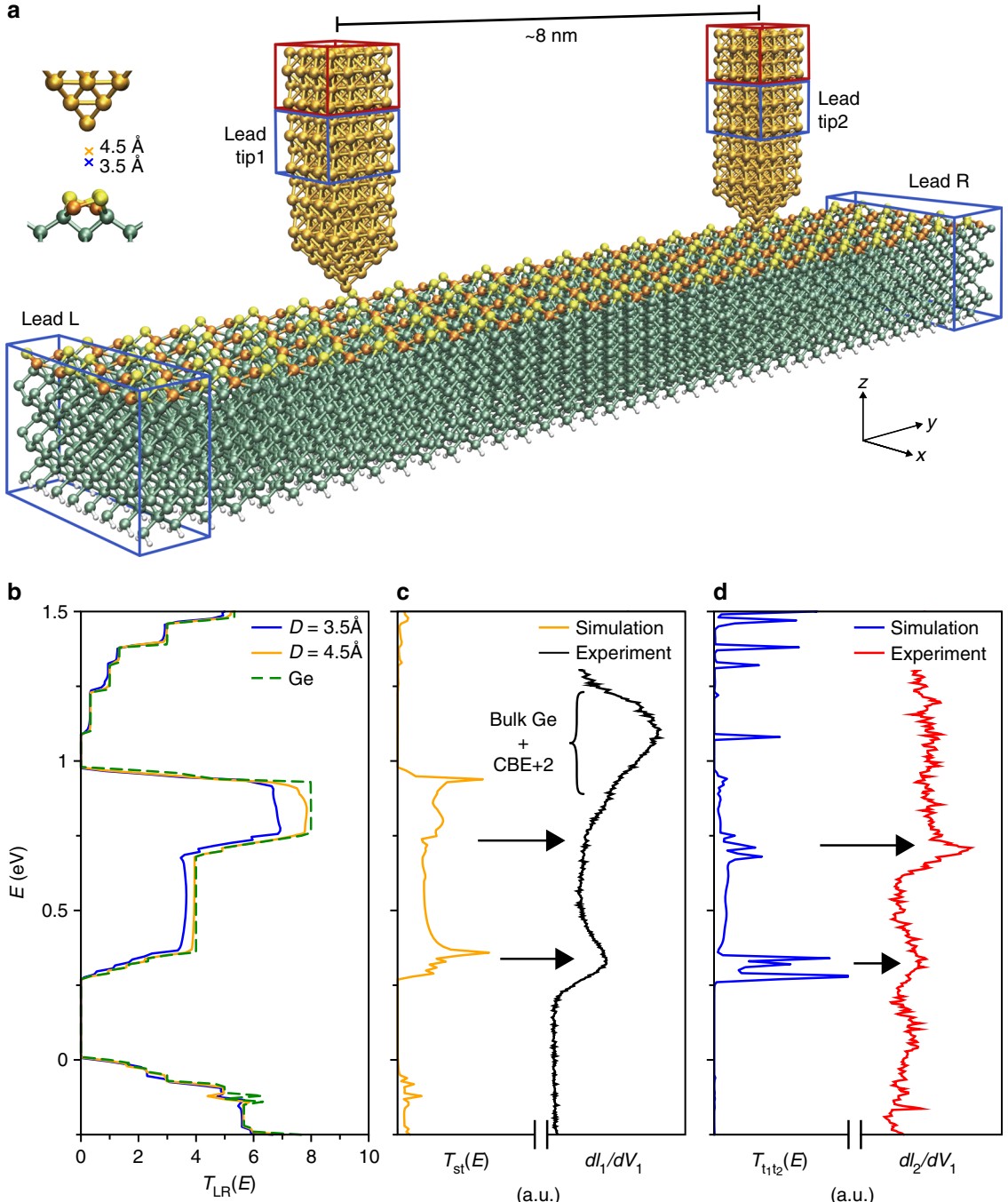

**Fig. 4** First-principles transport simulations for the two-probe experiments. **a** Representation of the four-terminal setup. The electrode regions are highlighted by blue boxes, two of them located at each Ge(001)-c(4 × 2) slab terminations (leads left and right) and the other two at each Au model tip (leads tip1 and tip2). The 50 Ge atoms closest to each tip were allowed to fully relax (Supplementary Fig. 8). **b** Left-to-right transmission function with a single tip, i.e., three-terminal setup. Pristine Ge slab transmission is included in dashed green as a reference. **c** Comparison between the experimental $-dI_1/dV_1$ spectrum and the calculated surface-to-tip transmission at zero bias for $D = 4.5$ Å (single-tip setup). **d** Comparison between the experimental $dI_2/dV_1$ spectrum and the tip-to-tip transmission function calculated for $D_1 = D_2 = 3.5$ Å with the setup represented in panel **a**. In all calculations the Ge slab valence band edge has been used as a common energy reference

energies (Fig. 4c); however, the bulk states are absent in our three-terminal setup. The intensity of the CBE + 1 peak depends strongly on the tip-surface distance (Supplementary Fig. 11 and Fig. 19). For the large tip-to-surface distances that mimic, the source probe STM/STS experimental conditions, $T_{st}(E)$ around the CBE + 1 resonance energy is relatively low. The ultimate reason for the low CBE + 1 peak intensity at these large tip-to-surface separations is not completely clear. However, the wave functions corresponding to the CBE + 1 bands present a strong phase modulation between neighboring dimers (Supplementary Fig. 18). Therefore, an s-symmetry tip wave function in the tunneling limit is expected to couple weakly to this band[55], which will contribute to further reduce the signal from this peak as the tip-to-surface distance is increased. Again, this corresponds to the experimental $-dI_1/dV_1$ where a hardly visible elbow near the CBE + 1 resonance can be identified in the 0.7–1.1 eV range.

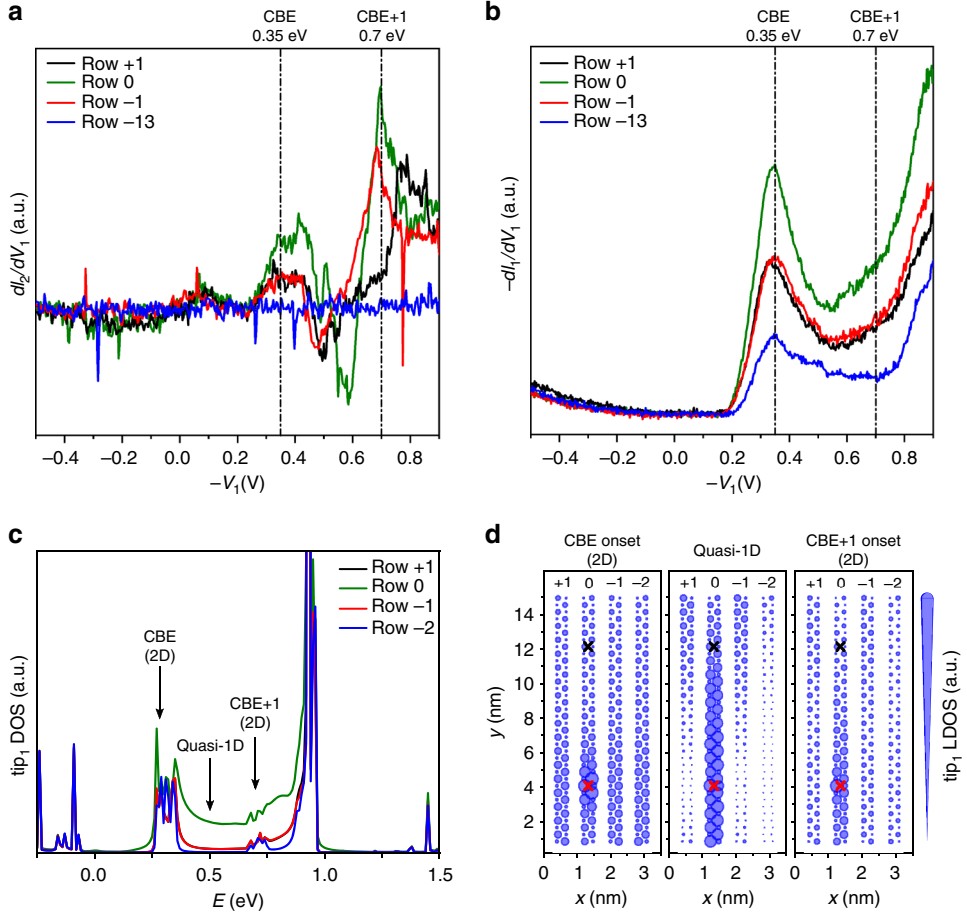

**Fig. 5** Electronic transport along neighboring Ge dimer rows. **a** Planar transconductance $dI_2/dV_1$ scanning tunneling spectroscopy (STS) signals as a function of tip1 voltage obtained for tip1 (source probe) located at different Ge dimer rows with respect to tip2 (drain probe). The corresponding separation across reconstruction rows are indicated on the label (0 marks the same reconstruction row). During acquisition of data the sample was grounded and the tunneling contact resistance of the tip2-sample junction was established to be ~50 MΩ and kept constant. Tip1-sample distance was established in all cases at $Z_0$ defined by $I = 20$ pA and $V = -0.5$ V. The lateral distance between probes was about 31 nm in each case. **b** Corresponding vertical $-dI_1/dV_1$ STS signals registered simultaneously with the data presented in **a**. The blue curves presented in **a** and **b** show reference data registered by tip1 positioned 13 reconstruction rows apart from tip2. **c** Density of scattering states incoming from tip1 (tip1-resolved DOS) projected on each of the four Ge dimer rows defining the simulation cell (for tip-to-surface distances $D_1 = D_2 = 4.5$ Å). **d** From left-to-right, the panels show the tip1-resolved local density of states obtained at 0.28 eV (conduction band edge(CBE) onset), at 0.5 eV (quasi-1D region), and at 0.72 eV (CBE + 1 onset), respectively. The radius of the circle centered at each atom is proportional to the calculated lead-resolved DOS. The lateral position of tip1 (tip2) apex atom is marked with red (black) crosses. In all calculations, the Ge slab valence band edge has been used as a common energy reference

Next, to simulate the 2P-STS experiments, a second metallic tip was introduced 8.0 nm apart from the first one on the same Ge dimer row (Fig. 4a). The simulated tip-to-tip transmission $T_{t1t2}(E)$ for $D_1 = D_2 = 3.5$ Å (Fig. 4d) reveals clear features around 0.3 eV and 0.7 eV. This is in remarkable agreement with the experimental $dI_2/dV_1$ spectra in Fig. 2c. Therefore, in a planar $dI_2/dV_1$ transconductance spectrum, the resonances observed at 0.35 eV and 0.7 eV can be assigned to the opening of two transport channels along the Ge dimer row. These channels are related to the corresponding CBE and CBE + 1 bands predicted from DFT calculations (Fig. 1a) and also detected by the FT-STS analysis (Fig. 3c). As expected, the computed $T_{t1t2}(E)$ curves are also strongly dependent on the tip-to-surface distance. Similar to the simulated single-tip STS, a clear CBE + 1 resonance only appears for relatively short distances $D \leq 3.5$ Å. This corresponds well with the actual experimental situation in which the drain probe forms a low-resistance contact to the surface. The experimental observation of a weak CBE + 2 resonance in the planar $dI_2/dV_1$ setup can be explained by the presence of Ge bulk

electronic states for energies above 0.9 eV. As indicated above, the opening of this channel for scattering into bulk is likely to efficiently reduce the lifetime of the electrons traveling along the Ge dimer row at those high energies, thus hindering their collection by the drain probe.

**Transport directionality of the surface states**. To shed more light on the relation between 2P-STS data and electronic structure of Ge(001)-c(4 × 2) surface, we discuss an experiment where the probes are shifted between consecutive reconstruction rows (Fig. 5, Supplementary Fig. 3). We followed the same methodology as used in the case of 2P-STS data from Fig. 2: tip2 was kept in tunneling conditions with low resistance (~ 50 MΩ) over a chosen Ge dimer row, whereas tip1 operating in high resistance tunneling conditions ($Z_0$ determined by − 0.5 V, 20 pA) was placed on consecutive Ge dimer rows. Fig. 5a presents transconductance $dI_2/dV_1$ spectra obtained for three consecutive rows with the central one (row 0) being occupied by tip2. Importantly, both CBE and CBE + 1 resonances are preserved

when probes are separated by a single reconstruction row. The CBE resonance is in this case strongly reduced by more than 50% in its intensity, whereas CBE + 1 is decreased only by about 20%. This trend continues for increasing number of separation rows. CBE resonance is not registered for three, while CBE + 1 is still observed even for seven reconstruction rows apart from tip2 position (see Supplementary Fig. 3). Interestingly, we also observe variations in the CBE and CBE + 1 resonance intensities of the simultaneously registered $-dI_1/dV_1$ spectra (Fig. 5b). The ultimate reason for this $dI_1/dV_1$ dependence is not clear. For example, at low energies one could invoke effects related to the interference between incoming waves from tip1 and those scattered by tip2. However, at 30 nm tip-to-tip distance, this effect should be rather small as our previous discussion demonstrates. In any case, in close agreement with the data already shown in the Two-probe STS experiment section, the registered $I_2$ is still in the range of 10% of $I_1$.

To interpret the presented 2P-STS data, we need to better understand the properties of the Ge(001) surface states. Interestingly, the analysis of the energy variation of the surface states over the whole Brillouin zone (see discussion in Supplementary Note 9) reveals a larger 2D character within a small energy window of ~ 100 meV right after the onset of each surface band (CBE and CBE + 1). In contrast, for the energy range in between (from ~ 0.36 to ~ 0.66 eV in Fig. 1a) the dispersion has a strong 1D character (Supplementary Fig. 20). In order to visualize how these two dimensionality crossovers affect electron propagation in these three energy ranges, we calculated the energy-dependent lead-resolved DOS and LDOS[56] projected on the Ge dimer rows. The lead-resolved DOS depicts how the electrons originating at a given lead (tip) distribute along the sample. Fig. 5c presents the tip1-DOS projected at three different reconstruction rows used in the simulation (rows − 1 and + 1 are similar due to symmetry). The lateral distributions of LDOS originating from tip1 for chosen energies are depicted in Fig. 5d. As expected, for energies corresponding to the onsets of the two surface bands (CBE and CBE + 1), the DOS is comparable at each of the reconstruction rows. Thus, the injected electrons at these energies have a more 2D character (see Fig. 5d). On the other hand, for energies from ~ 0.36 to ~ 0.66 eV, we observe strong anisotropy in the DOS, which in this case is mainly distributed over a single Ge dimer row. The latter result reflects the small transverse (compared with parallel) group velocity of electrons at these intermediate energies, i.e., it confirms the strong quasi-1D character of the transport in the Ge(001)-c(4 × 2) surface away from the onsets of the surface bands.

The experimental data from Fig. 5a, b can thus be interpreted in the following way. 2P-STS data captures CBE and CBE + 1 resonances, which are related to the opening of two quasi-1D transport channels along a single Ge dimer row. Interestingly, at the energies of the CBE and CBE + 1 resonances (band onsets), the corresponding bands have a non-negligible 2D character; thus, even though the transport is highly anisotropic, the corresponding resonances are expected to be registered also on the neighboring rows of the reconstruction. In addition, even if transport would be mostly coherent, we should always expect increased signal of the $dI_2/dV_1$ at the band minima (CBE and CBE + 1 resonances), as they get populated by the fraction of electrons inelastically scattered within each surface band[43]. This easily explains the lack of direct proportionality between $-dI_1/dV_1$ and $dI_2/dV_1$.

## Discussion
Using 2P-STM/STS instrumentation with tip separation distances down to 30 nm, we performed 2P-STS planar measurements along a single dimer row on the bare Ge(001)-c (4 × 2) surface. A remarkable agreement was found between the calculated electronic transmission and the experimental $dI_2/dV_1$ transconductance spectra, allowing to interpret the results in terms of the surface band structure of the system. Two transconductance resonances were identified and assigned to two quasi-1D transport channels existing along each of the surface Ge dimer rows. This picture was corroborated by an analysis of interference patterns near step-edges using single-probe STM/STS, as well as by first-principles transport simulations. Application of the FT-STS method allowed us to reconstruct the dispersion of electronic surface states. A striking feature of these surface-propagating electrons is that their coherence is preserved at distances up to 50 nm. The identified coherent nature of the surface channels opens the possibility to control the electronic transport along Ge dimer rows by engineering quantum interference[57,58], e.g., with defects, adsorbates, or mechanically operated probes. From a more general perspective, the presented protocol can be used to characterize transport at the nanoscale in planar atomic-scale devices and 2D materials grown on surfaces. In contrast to standard metal contacts, e.g., fabricated by lithographic techniques, the use of 2P-STM enables precise adjustment of individual atomic contacts and their resistances. This additional level of control helps to access the system's intrinsic transport properties, disentangling them from those of the contacts and leads. Finally, to provide complete control over structural details of the atomic contacts, our 2P-STM approach can be easily combined with STM tip apex functionalization protocols.

## Methods
**Experimental details**. The experiments were carried out in the ultrahigh vacuum system equipped with the LT-Nanoprobe low-temperature four-probe STM[27,28]. The experiments were carried out at cryogenic temperature of around 4.5 K with electrochemically etched platinum-iridium alloy wires used as probes. Before the experiment, the tips were sharpened by focused ion beam method. Initial coarse positioning of STM probes is performed using scanning electron microscope. For single-probe STM/STS $dI/dV$ measurements, we used standard experimental design based on application of a lock-in amplifier (20 mV peak to peak, 550 Hz). The 2P-STS data were obtained by two lock-in amplifiers setup (see Supplementary Notes 1 and 4 for details). STM and $dI/dV$ map images were only flattened with the use of SPIP software. All STS data present raw (as collected) points.

The Ge(001) samples (2 × 10 mm$^2$, TBL Kelpin Crystals, 0.5 mm thickness, undoped) were prepared in a standard manner by series of 1 keV Ar$^+$ sputtering for 15 min with the sample kept at 1040 K[29].

**Simulation details**. First-principles DFT calculations were performed with the SIESTA package[59,60]. Transport properties were computed with NEGF techniques as implemented in TranSIESTA[56,61], which allows for simulations with open boundary conditions and multi-terminal configurations. Due to the complex systems explored experimentally and the long screening lengths that characterize semiconducting systems, realistic simulations should comprise many atoms (in our case up to 5000 atoms). It is therefore critical to find a suitable compromise between different computational parameters that allows for a good description of the physics without increasing too much the computational cost. A description of the employed simulation parameters as well as all the consistency verifications are provided in Supplementary Note 5. All transmission functions presented in Fig. 4 are averaged over transversal $k$-points and evaluated at zero bias. Taking into account that for a bare Ge(001) surface prepared under vacuum conditions, the Fermi level is usually pinned at the valence band edge (VBE)[43,44,62,63], we present all our calculated results with respect to the pristine Ge slab VBE energy. The valence band electronic states at the Fermi energy effectively screen the electric potential differences resulted from different electron work functions between metallic tips and a Ge surface and thus minimize effects typically present on other semiconducting surfaces during STM/STS experiments. This was one of the practical reasons to use the Ge(001)-c(4 × 2) surface as the model system for our 2P-STS experimental and theoretical analysis.

## Data availability
The data that support the findings of this study are available from the corresponding authors on request.

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

## Acknowledgements

We thank T. Skeren, A. Fuhrer (IBM Zurich Research Laboratory), B. Jany, and F. Krok (Jagiellonian University) for focused ion beam tip preparation and N. Papior (Technical University of Denmark) for helpful discussions regarding TranSIESTA. This work was supported by the FP7 FET-ICT "Planar Atomic and Molecular Scale devices" (PAMS) project (funded by the European Commission under contract number 610446), the Polish Ministry for Science and Higher Education from financial resources for science in 2013–2017 granted for an international co-financed project (contract number 2913/7.PR/2013/2) and the Spanish Ministerio de Economía y Competitividad (MINECO) (Grant Numbers MAT2016-78293-C6-4-R and FIS2017-83780-P). M.K. acknowledges financial support received from the Polish Ministry of Science and Higher Education, contract number 0341/IP3/2016/74. Part of the work was conducted at the Center for Nanophase Materials Sciences (CNMS), which is a DOE Office of Science User Facility.

## Author contributions

M.K. conceived, designed, performed, and analyzed the experiment. J.L., R.Z., S.G. and M.S. supported experiments and data analysis. P.B. performed the calculations. P.B., H.K., A.G.-L., N.L., T.F., C.J. and D.S.P. provided theoretical analysis and interpretations. M.S., D.S.P. and C.J. supervised the research. The manuscript was written by M.K., P.B. and D.S.P. All the authors discussed the results and revised the manuscript.

## Additional information

**Competing interests:** The authors declare no competing interests.

