## [Peer Review File · Nature Communications]

Reviewers' comments:

Reviewer #1 (Remarks to the Author):

The manuscript reports on a joint experimental/computational study of coherent transport in atomic-scale structures (dimer rows on Ge(100)), matching high quality two-probe STM/STS data with large scale ab initio simulations of quantum transport.

To my understading, the crucial aspect of the paper is in use of (beyond) state-of-the-art experimental techniques combined with top-level state-of-the-art ab initio calculations (based on DFT and non-equilibrium green's functions). The interplay of the two approaches is very well documented in a number of aspects (both in the main text and SI), and shows that a quantitative agreement can be reached. In turn, the paper represents a strong validation of the study protocol proposed by the authors.

The combined method is applied to the (100) surface of Ge, a prototype system in this respect, and reveals a quasi 1D nature of transport along the reconstructed dimer rows. If any, I would suggest the Authors to extend the introduction and system motivation by describing other possible scenarios (possibly beyond Ge 100) where the current approach can be used and exploited.

Overall, in view of the above, I do recommend publication of the manuscript.

Reviewer #2 (Remarks to the Author):

This manuscript reports on a demonstration of novel transport measurements by two-probe scanning tunneling spectroscopy (2P-STs). 2P-STs utilizes two metal tips, positioned over a target sample with a distance shorter than the coherence length of electrons in the system. Electrons (or a current) injected by one tip propagate through the sample and are collected by the other tip. This method was proposed as a way to evaluate the Green function of an electron in the system more than two decades ago, but it had not been experimentally demonstrated. The authors chose a germanium atom dimer row forming on the Ge(001) surface as a test system. The row acts as a 1D electron propagator with anisotropic band dispersion emerging within the Ge bulk band gap, which was confirmed by conventional STM observation of standing waves. The transconductance spectra were interpreted with the aid of DFT band and transport calculations. This manuscript includes possibly technological groundbreaking results in the research field of SPM, and maturation of this technique has the potential for applications to various material transport property measurements.

Since this technique is practically new, the authors need to be very careful to establish credibility of this method by extremely careful experimental design and thorough interpretation of data. At a glance, this technique seems to have a technical defect that can be immediately seen by drawing an equivalent circuit diagram to the experimental instrument and conditions: The tip2 current may be actually recorded by approximately three orders of magnitude smaller than V_1 . Experimental test results included in this manuscript are not enough to dispel my doubt about this technique. Even if the transconductance measurement is correctly done, there are still several points remaining unclear regarding the band calculations and the interpretation of their data. Finally, in my opinion, the manuscript is hard to read as the reader must make continual reference to the supplemental material. Therefore, I must conclude that the manuscript in its present form is not suitable for publication in Nature Communications. I give detailed questions and comments below.

1. The central question that I have is whether I_2 or dI_2/dV_1 signals detected by the tip2 throughout the experiment really include information from the band structure. Figure A shows schematics of the experimental setup read from the manuscript and its two possible equivalent circuit diagrams. R_1 and R_2 are resistances of tunnel gap at the tip1 and tip2, respectively. R_3 is a resistance between surface state and bulk state. The authors interpreted their transconductance measurement data based on the model circuit in (b), according to Fig. 1c. In this model, electrons injected from the tip1 to a dimer propagate through a single dimer row. Then, they are collected by the tip2. In spite of the fact that electrons go through two tunnel gaps, electrons experience a potential drop only at the tunnel gap2. On the other hand, the same experiment is possibly

interpreted as a model circuit (c). In this model, electrons experience potential drops both at tunnel gaps 1 and 2. Fig. 1e shows that approximately $I_1 = 1$ nA and $I_2 = 0.1$ nA at 1.5 V. This indicates that R_1 was actually set to 1.5 G Ω , instead of "a junction resistance larger than 100 G Ω " as written in line 115. R_2 was 50 M Ω from the text. I_3 at $R_3 = 0.9$ nA, because $I_2 = 0.1$ nA. Thus, $R_3 = 50/9 = 5.6$ M Ω , and the combined resistance of R_2 and R_3 is 1505 M Ω . This immediately gives the potential drop at $R_1 = 1.5 * 1500/1505 = 1.495$ V and that at $R_2 = 5$ mV. Therefore, while dI_1/dV_1 spectrum covers the π^* states of the Ge dimers in almost the same energy as applied bias V_1 , dI_2/dV_1 measures only up to 5 mV, which is too small to detect any of the π^* states. The authors should experimentally check which model is more appropriate by changing the value of R_1 and by observing the dI_2/dV_1 spectral shape.

2. The band diagram in Fig. 1a shows an unusual feature of unoccupied dangling bond (π^*) band splitting. There are four bands in the unoccupied state, because their band calculation was done by a supercell with four Ge dimers. If the same calculation is done by the $c(4 \times 2)$ primitive cell with two dimers, only two bands are obtained, as published in ref 34. The extension of the cell size should result in extra degenerate bands or band folding because all dimers should have the equivalent structure. Have the authors considered that the band splitting may be caused by an artifact in their calculation? Or do the authors have a good physical reason for this band splitting?

3. Related to the previous question, does the band structure experimentally obtained from the standing waves in Fig. 2e really show split bands? It can hardly be seen in the figure because the calculated bands (blue curves) are superimposed. Taking cross section profiles along the wave vector at a few energies should help to find two bands, if the bands are really split.

4. Line 190: "The dispersion of these bands agrees also with previously reported single-probe STS studies [33, 34, 43] and with a very recent angle resolved two-photon photoelectron spectroscopy experiment [36]." Again, neither of cited papers showed a split π^* band.

5. To plot band dispersions in Fig. 2e from standing waves, the authors used cosine waves with two wave vectors (SI Section S5). However, three k vectors are plotted at 0.65 eV and 0.7 eV. How did they do this? I would like to see dI/dV cross section profiles along the dimer row and their fitting curves. In ref 33 and ref 34, previous fitting to standing waves were done with only one k vector. How do the authors explain this inconsistency?

6. Line 114: "Marked as tip1 in Fig.1" should be "Fig.1d."

7. While line 114 reads "the first STM probe is kept in a tunneling regime with a junction resistance larger than 100 G Ω ," line 130 reads "Tip1 (blue circle) was kept in a tunneling condition ($I=10$ pA, $V_{\text{sample}}=-0.5$ V). This corresponds to 50 G Ω . Furthermore, line 90 (caption for Fig. 1) reads "Filled-state STM images ($I=20$ pA, $V_{\text{sample}}=-0.5$ V) obtained prior to the 2P-STs experiment." This tunneling parameter corresponds to 25 G Ω . Furthermore, I pointed out in my comment 1 that the tunnel resistance is 1.5 G Ω in Fig. 1e. These inconsistencies are very confusing.

8. Where were the tip1 and tip2 positioned exactly? If the tips are assumed to be positioned at the center of blue and white circles in Fig.1d, the tips were over the up atoms of the Ge dimers because the image was taken in filled state. But the transconductance measurement was carried out over the π^* state, which is predominantly distributed over the down atoms of the dimers. Is there any specific reason to have chosen the up atoms?

9. The dI_2/dV_1 spectrum in Fig.1f shows a small peak at 0.1 V, which has almost the same magnitude as the peak at 0.35 V. It also appears in Figs. S1 and S2. Do the authors have any comment on this peak?

10. A schematic of the Brillouin zone would help the reader to understand the band structure in Fig.1a.

11. Line 234: "In our self-consistent multi-terminal treatment, we considered a model system composed of a six-layer Ge(001)- $c(4 \times 2)$ slab contacted..." The authors used a 12-layer slab, didn't they?

12. Line 263: "Furthermore, the wave functions corresponding to the CBE+1 bands present a strong phase modulation between neighboring dimers (Fig. S14). Therefore, an s-symmetry tip wave function in the tunneling limit is expected to couple weakly to this band [44], which will contribute to further reduce the signal from this peak as the tip-to-surface distance is increased." I cannot agree with this statement. Fig. S13(c) shows that the main orbital producing the CBE+1 band is P_z , which is more likely to interact with the s orbital of the tip than with the other orbitals. How is the coupling of wavefunctions between the tip and the dimer influenced by the phase of the wavefunction of an adjacent dimer?

13. Among simulation data in Fig. 3 b-d, there is consistency in specific energies (0.3 eV, 0.7 eV and 0.9 eV) at which characteristic transmission probabilities appear. The authors claim that this is because all these features originate from the opening of new conduction channels such as CBE, CBE+1 and CBE+2 (CBE+2 is rather resonant with bulk, though). Moreover, they found remarkable agreement between the experimental dI_2/dV_1 spectrum and simulation in Fig. 3d, and

accordingly the peak at 0.7 V was assigned to CBE+1. On the other hand, the experimental dI_1/dV_1 spectrum features were similarly assigned to CBE, CBE+1 and CBE+2, even though their energies show significant deviation from the simulated energies. I think this is not a reasonable interpretation.

14. Fig. S2: I think this is a very important test to understand the 1D nature of the dimer row and verify the credibility of the transconductance measurement. Can the authors provide a more systematic measurement for different dimer rows such as row0, row-1, row-2, row-3, ...? Why does the magnitudes of the dI_1/dV_1 spectrum vary with the position of the tip1 over different dimer rows?

15. Fig. S3: The authors claim that the coherence length of electrons in the π^* state is about 50 nm. dI_2/dV_1 measurement should be done with the tip1-tip2 separation over 50 nm to support this and also to see the performance of 2P-STs. Moreover, the shape of dI_2/dV_1 spectra in Fig. S3c do not look the same as that in Fig. 1f, which makes the credibility of measurement weaker.

Figure caption

Figure A: Experimental setup for 2P-STs (a) and two possible equivalent circuit diagrams (b) and (c).

Reviewer #3 (Remarks to the Author):

The authors present unique experimental current transport measurements at a Ge surface on the nanoscale. However, I have some concerns regarding the interpretation of the data, which should be clarified by the authors:

1. Fermi level of the bulk Ge, surface band bending, tip induced band bending:

In Fig. 1c the Fermi level in the Ge bulk is shown to be located at the valence band, which corresponds to degenerate p doping of the sample. On the other hand the authors mention in the methods section that the sample is undoped, which corresponds to the Ge bulk Fermi level located around mid-gap. In my opinion the interpretation of the results will change depending on this. Further it seems that things like surface band bending and tip induced band bending are not considered in the interpretation of the data.

2. LDOS oscillations at a step edge, coherent transport:

The authors present dI/dV oscillations close to a step edge. The authors use these results to assign the current transport as coherent. However, this assignment is not made from the transport measurements itself, but very indirectly via the observation of the oscillations. Due to this, I would strongly recommend to change the title of the paper from Coherent transport... to Current transport... Moreover, (as the authors mention) these results have already been obtained and published by two groups more than ten years ago. Therefore, I would consider these results more for the supplementary information section than for the main text. The assignment of the transport as coherent is based on these well-known result and should not be mentioned in the title as main result of the current paper.

3. Fig. S2

I find the data in Fig. S2 much more interesting than the already known LDOS oscillations at a step edge and would propose to include this into the main text. However, these data should be discussed in more depth: Why is the dI_1/dV_1 decreasing from the black trace to the blue trace? Should the dI_2/dV_1 data be scaled by the peak height of the dI_1/dV_1 data as reference? In this case the red trace would become as high as the black one. Why is the CBE +1 not visible in the dI_1/dV_1 spectrum (This is also the case in Fig 1f)? The data may be also interpreted in a much simpler way as an anisotropic conductivity of the surface states with higher conductivity along the dimer rows and lower conductivity perpendicular to the dimer rows.

In my opinion, the manuscript might be suitable for publication in Nature Comm. if the authors can dispel my concerns regarding the interpretation of the data.

Reviewer #1 (Remarks to the Author):

The manuscript reports on a joint experimental/computational study of coherent transport in atomic-scale structures (dimer rows on Ge(100)), matching high quality two-probe STM/STS data with large scale ab initio simulations of quantum transport.

To my understanding, the crucial aspect of the paper is in use of (beyond) state-of-the-art experimental techniques combined with top-level state-of-the-art ab initio calculations (based on DFT and non-equilibrium green's functions). The interplay of the two approaches is very well documented in a number of aspects (both in the main text and SI), and shows that a quantitative agreement can be reached. In turn, the paper represents a strong validation of the study protocol proposed by the authors.

The combined method is applied to the (100) surface of Ge, a prototype system in this respect, and reveals a quasi 1D nature of transport along the reconstructed dimer rows.

If any, I would suggest the Authors to extend the introduction and system motivation by describing other possible scenarios (possibly beyond Ge 100) where the current approach can be used and exploited.

Overall, in view of the above, I do recommend publication of the manuscript.

We thank the reviewer for his/her positive comments on our work and for recommending publication. According to the suggestion we have added the following part to our introduction showing two recently attractive topics, where our two-probe STM methodology could be directly applied.

The last part of introduction paragraph now reads (previous line 74):

“Besides this exploration of prototypical atomic circuits, our measurement protocol provides a general tool to explore in-plane electronic transport applicable to high interest research fields including for example engineered 2D systems [22, 33-38] or materials with topological electronic states [39-42]”.

Reviewer #2 (Remarks to the Author):

This manuscript reports on a demonstration of novel transport measurements by two-probe scanning tunneling spectroscopy (2P-STs). 2P-STs utilizes two metal tips, positioned over a target sample with a distance shorter than the coherence length of electrons in the system. Electrons (or a current) injected by one tip propagate through the sample and are collected by the other tip. This method was proposed as a way to evaluate the Green function of an electron in the system more than two decades ago, but it had not been experimentally demonstrated. The authors chose a germanium atom dimer row forming on the Ge(001) surface as a test system. The row acts as a 1D electron propagator with anisotropic band dispersion emerging within the Ge bulk band

gap, which was confirmed by conventional STM observation of standing waves. The transconductance spectra were interpreted with the aid of DFT band and transport calculations. This manuscript includes possibly technological groundbreaking results in the research field of SPM, and maturation of this technique has the potential for applications to various material transport property measurements.

We thank the reviewer for finding our results interesting and for recognizing their potential groundbreaking nature for the multiprobe STM field.

Since this technique is practically new, the authors need to be very careful to establish credibility of this method by extremely careful experimental design and thorough interpretation of data. At a glance, this technique seems to have a technical defect that can be immediately seen by drawing an equivalent circuit diagram to the experimental instrument and conditions: The tip2 current may be actually recorded by approximately three orders of magnitude smaller than V_1 . Experimental test results included in this manuscript are not enough to dispel my doubt about this technique.

We thank the reviewer for expressing his/her constructive criticism about our novel approach. This opinion motivated us to more strongly underline the methodology behind the experimental and theoretical analysis of the data presented in our manuscript. Based on the reviewer's comments we decided to include an additional section in SI (now S2). This section includes tip-surface contact resistance determination methodology used in this work. We have also included additional data on the dependence of transport on the relative positions of the STM tips along the direction perpendicular to the Ge-dimer wires in a novel part of our manuscript (last section). We believe that our detailed response together with these additional results supporting our main conclusions will dispel the raised doubts.

Even if the transconductance measurement is correctly done, there are still several points remaining unclear regarding the band calculations and the interpretation of their data. Finally, in my opinion, the manuscript is hard to read as the reader must make continual reference to the supplemental material. Therefore, I must conclude that the manuscript in its present form is not suitable for publication in Nature Communications. I give detailed questions and comments below.

We thank the reviewer for his/her opinion. We agree that the points raised were not clear enough in the previous version of the manuscript. Regarding the structure of the manuscript, it is always difficult to find the right balance between the need to provide a precise and self-contained account of the investigation and the amount of results presented in the main text. As pointed out by the referee, our manuscript presents a novel technique and, in order to be convincing, we think that it is necessary to present a very detailed account of our results. We think that the supporting information is the right place to present all these additional results. We have opted to include in the supplementary material relevant measurements and theoretical analysis made to support our claims as well as a detailed description of the methodology, while keeping the main message in the manuscript. Nevertheless, we agree with the reviewer that some parts contained too many references to the SI and we have tried to reduce accordingly to help the reading flow. We hope that our detailed answers for all the

comments and the revised and improved version of the manuscript will be positively reconsidered.

1. The central question that I have is whether I_2 or dI_2/dV_1 signals detected by the tip2 throughout the experiment really include information from the band structure.

We agree with the reviewer that the collected two-probe STS data themselves do not prove that the resonances are related to the electronic structure of the nanostructure. Please note that the same problem exists in all STM experiments, which *in principle* could be mimicked by a proper classical circuit of (nonlinear) resistors. Thus, in the originally submitted work we supplemented our two-probe experimental data with the well-established single-probe STS analysis performed on the same system proving the coherent nature of transport (section two of the manuscript), and with state-of-the-art multi-terminal transport calculations performed using non-equilibrium Green's functions (section three). In the revised version we include also additional position-dependent two-probe STS data (new section **Surface states transport dimensionality**).

Figure A shows schematics of the experimental setup read from the manuscript and its two possible equivalent circuit diagrams. R1 and R2 are resistances of tunnel gap at the tip1 and tip2, respectively. R3 is a resistance between surface state and bulk state. The authors interpreted their transconductance measurement data based on the model circuit in (b), according to Fig. 1c. In this model, electrons injected from the tip1 to a dimer propagate through a single dimer row. Then, they are collected by the tip2. In spite of the fact that electrons go through two tunnel gaps, electrons experience a potential drop only at the tunnel gap2.

We apologize because there must be a misunderstanding here. It comes probably from the lack of a proper explanation in the caption of **Fig.1** in our original text. Nowhere in the manuscript do we claim or intend to claim that the Reviewer's circuit (b) is an appropriate model of our experiment. While we think that it is potentially dangerous and might be misleading to try to push too far the analogy with a classical circuit, we tend to agree with the referee that our experimental set up is better represented by circuit (c). Nevertheless, **Fig. 1c** in our manuscript depicts the nanoscale scattering region by showing a scheme of the level alignment with the applied bias at *tip1*. Both the surface and *tip2* are grounded (I_2 current is measured with no bias voltage applied on the *tip2*-sample junction) and in order to interpret correctly **Fig. 1c** it must be taken into account that the Fermi energy of the Ge(001) surface is pinned at the top of the bulk Ge valence band [46-49]. This point was mentioned only in the simulations description at the **Methods** section in the original manuscript. Therefore, **Fig. 1c** intends to represent the situation where the chemical potential of both Ge and *tip2* are aligned with no bias difference applied. This point is emphasized in the new version of the **Fig. 1** caption to avoid confusion, where the following sentence has been appended:

“Notice that the Fermi energy of Ge(001) is known to be pinned at the top of the Ge bulk valence band [46-49]. Thus, in our scheme the chemical potentials of tip2 and the surface are aligned.”

Additionally, our scheme emphasizes the mechanism of electronic injection through the Ge surface states. Considering that electrons propagate coherently through the surface (disregarding sources of inelastic scattering), an injected electron having an energy in the range between CBE up to CBE+1, cannot decay to bulk, but can be collected by *tip2* whose chemical potential is lower in energy.

As our methodology resembles the ballistic-electron emission microscopy (BEEM) concept, we have added the corresponding sentence to **Fig.1** caption (previous line 89):

“The experimental design resembles ballistic-electron emission microscopy concept [43-45] with tip2 acting as a collector.”

We kindly ask the reviewer to consider two seminal references [43,44] and particularly to take into account the mechanism of BEEM signal detection when reading the following part of our answer.

On the other hand, the same experiment is possibly interpreted as a model circuit (c). In this model, electrons experience potential drops both at tunnel gaps1 and 2.

Fig. 1e shows that approximately $I_1 = 1$ nA and $I_2 = 0.1$ nA at 1.5 V. This indicates that R_1 was actually set to 1.5 G Ω , instead of “a junction resistance larger than 100 G Ω ” as written in line 115. R_2 was 50 M Ω from the text. I_3 at $R_3 = 0.9$ nA, because $I_2 = 0.1$ nA. Thus, $R_3 = 50/9 = 5.6$ M Ω , and the combined resistance of R_2 and R_3 is 1505 M Ω . This immediately gives the potential drop at $R_1 = 1.5 * 1500/1505 = 1.495$ V and that at $R_2 = 5$ mV.

We thank the reviewer for raising this point as it evidences that we missed an explanation of how the mentioned junction resistances were defined. Since the STS I-V curves on Ge(001) surface are strongly non-linear and due to high current densities for larger biases, we can measure directly all the resistances only around Fermi energy, where one expects linear I-V characteristics of all resistances. Those measurements are now included in the section 2 from SI as shown below.

“S2. Probe to sample contacts

FIG. S1. Characterization of the contact between PtIr STM tip and the Ge(001) surface for different tip-surface distances at 4.5 K. Graphs in (a-c) present *the same sets* of data at three current regimes. For each tip-sample distance we present two sets of raw data. The I-V characteristics around Fermi energy show linear dependence due to the Ge(001) surface Fermi level pinning effect¹⁻⁴. Starting from high resistance tunneling conditions $Z_0(-0.5\text{ V}, 20\text{ pA}$, black points) the tip is progressively approached towards the surface by 0.1 (orange), 0.2 (green), 0.3 (blue), 0.4 (pink) and 10 nm (red), what changes the corresponding junction resistances from above 100 G Ω (pure tunneling conditions) to about 50 k Ω (saturated value of resistance reflecting multi-channel Ohmic contact). Note that standard two and four probe experiments are performed in the regime of Ohmic contacts, what hinders the understanding of the atomic-scale processes behind the electronic transport at the tip-sample junction. Also note that such a low resistance Ohmic contact requires a large contact area and a strong interaction between tip and surface that is likely to cause strong non-controllable modifications on the surface structure.”

Therefore, while dI_1/dV_1 spectrum covers the π^* states of the Ge dimers in almost the same energy as applied bias V_1 , dI_2/dV_1 measures only up to 5 mV, which is too small to detect any of the π^* states.

It is reassuring to find that the estimation based on a classical analogy (in spite of its limited applicability to the nanometer scale, where transport is primarily coherent as shown below) reaches to the conclusion that the bias drop at *tip2* is very small, which is fully consistent with the actual experimental situation.

However, it is instrumental to note that such negligible bias drop has nothing to do with the energies of the electrons being collected by *tip2*. The energies of those electrons are determined by the surface bands at which they are injected from *tip1*. Thus, the only limiting factor is the V_1 bias. For example, if the reasoning presented by the reviewer were applied to the referred BEEM experiments [43-45], the BEEM collector current could not be detected as the potential drop on the base-collector junction would be about two orders of magnitude lower than corresponding Schottky barrier heights.

In our experiment hot electrons injected into the unoccupied Ge surface states through tunneling junction 1 have a mean free path longer than the distance between probes, what is nicely demonstrated by our single probe step-edge reflection experiment. Thus, some electrons are captured by unoccupied states at *tip2* without energy loss (elastic transfer).

What could be somewhat misleading here is the presentation of dI_2/dV_1 spectra as a function of the *tip1*-sample bias. Even if we cannot directly probe the energy of electrons “arriving” to *tip2* we decide to keep the data presentation this way. Note that the similar notation is used by the BEEM community.

Finally, please note that in the literature there are many experiments proving that charge carriers emitted from STM tip into planar surface states of semiconductors can keep their coherent nature on distances of tens of nanometers. Particularly, the work by Sloan *et al.* [52] shows that ballistic holes emitted into surface states of Si(111) can induce desorption of toluene molecules at distances of about 30 nm from the injection point (at room temperature). Moreover, these authors show that the probability of this non-local manipulation is directly related to the electronic structure of Si(111) surface states (see Fig. 2 of their article). Thus, Sloan *et al.* conclude that holes injected from STM tip propagate coherently through the surface states before they induce molecule desorption events.

We decided to include the work by Sloan *et al.* as a reference in our manuscript (at the previous line 116):

“This junction plays the role of a source probe, injecting hot carriers [52] into the Ge electronic states of the selected single row.”

The authors should experimentally check which model is more appropriate by changing the value of R_1 and by observing the d_2/d_1 spectral shape.

We thank the reviewer for this suggestion. However, as *tip1*-sample junction plays the role of current source, i.e., injecting hot electrons into the system, it must have large tunneling resistance R_1 values. Larger currents driven through the Ge dimer rows could induce unwanted effects including oscillations of the Ge dimers, which will affect both the stability of the contacts and the transport properties along the Ge-dimer wires. In this work we tried to

use injected currents not exceeding 1 nA for the highest *tip1*-sample bias used. Variations of R_1 within this tunneling range do not affect the dI_2/dV_1 spectra. This was shown in the previous **Fig. S1** (now **Fig. S2a**), where the blue dI_2/dV_1 spectrum has about twice smaller R_1 resistance than the other two spectra. The dI_2/dV_1 spectral shape depends much more on R_2 contact resistance (see new **Fig. S2b**) and relative lateral positions of STM probes.

In the revised version we underline these differences in the new **Fig. S2**:

Fig. S2. (a) Set of three 2P-STs data obtained by probes positioned on the same Ge dimer row on $c(4 \times 2)$ reconstructed Ge(001) for probe-to-probe distance of 30 nm. Three pairs of simultaneously obtained vertical $-dI_1/dV_1$ and planar transconductance dI_2/dV_1 signals as a function of *tip1* voltage are shown by black, red and blue curves respectively. The resonances observed in the dI/dV characteristics at energies 0.35 eV, 0.7 eV and 1.1 eV are ascribed to surface conduction band edge (CBE), CBE+1 and CBE+2, respectively. Data were obtained with the application of the protocol with two lock-in amplifiers at the same experiment conditions as discussed in the main text (**Fig. 2c**), excluding closer *tip1*-sample Z_0 distance for blue spectra, which was defined by $V_{\text{sample}} = -0.5$ V and $I = 20$ pA (instead of 10 pA for other data). Note that about two times lower tunneling resistance of *tip1*-sample junction does not affect the general characteristics of transconductance signal. **(b)** Set of three 2P-STs transconductance data as a function of *tip1* voltage obtained by probes positioned on the same Ge dimer row on $c(4 \times 2)$ reconstructed Ge(001) for different values of *tip2*-surface low-bias resistance (see legend). *Tip1*-surface distance was defined by $V_{\text{sample}} = -0.5$ V and $I = 10$ pA. The relative distance between probes was 30 nm for black and 37 nm for red and blue spectra. Note that relative intensities of resonances in transconductance dI_2/dV_1 signals reflect similar trend to calculated spectra presented in **Fig. S11e,f**.

2. The band diagram in Fig. 1a shows an unusual feature of unoccupied dangling bond (π^*) band splitting. There are four bands in the unoccupied state, because their band calculation was done by a supercell with four Ge dimers. If the same calculation is done by the $c(4 \times 2)$ primitive cell with two dimers, only two bands are obtained, as published in ref 34. The extension of the cell size should result in extra degenerate bands or band folding because all dimers should have the equivalent structure. Have the authors considered that the band splitting may be caused by an artifact in their calculation? Or do the authors have a good physical reason for this band splitting?

We thank the reviewer for raising this question, allowing us to improve the text to avoid misinterpretations. Instead of taking a primitive cell, which as correctly pointed by the reviewer contains only two dimers, we rather prefer to use a supercell defined by orthogonal lattice vectors. While doubling the size of the cell, this choice permits to associate easily the band structure with the directions relevant to the experimental setup. In particular, the Γ -Y direction runs parallel to the Ge-dimer lines in the substrate (y-axis) and contains two slightly shifted replicas of each surface band (see Fig 1a), indicating the small dispersion of these quasi-1D surface bands along the direction perpendicular to the dimer lines. These are the duplicated theoretical bands appearing in Fig. 2e of our original manuscript and are particularly relevant since they give the maximum and minimum allowed energies for each surface band at a particular value of k_y , i.e, the bandwidth due to the perpendicular dispersion is directly given by their relative energy shift. Alternatively, and more relevant for our experimental STS spectra (see the discussion below), these two threshold bands provide the minimum (k_y^{\min}) and the maximum (k_y^{\max}) allowed values of k_y for each surface band (e.g., CBE) at a given energy. We agree with the referee that this is an important point that deserves to be better explained in our manuscript. We list below the changes made on the text to clarify this point:

In the caption of Fig. 1 (previously lines 80-81) now reads:

“(a) Calculated band structure of a twelve-layer Ge(001)- $c(4 \times 2)$ slab (inset presents the top view of the used unit cell, twice the size of the primitive cell, and the corresponding Brillouin zone).”

In the second paragraph of the “Electronic structure of the system” section (previously line 188) now reads:

“Starting at about 0.3 eV, the surface band corresponding to CBE presents two higher intensities corresponding to the minimum and maximum of the band dispersion along the perpendicular to the Ge-dimer wires, noted as #1 and #2 respectively. The energy positions and dispersions are in good agreement with the band structure calculations as shown in the left side of Fig. 3e where we present the computed k -resolved density of states (DOS, equation 3 in section S9 in SI), which confirms that a higher DOS is expected at the energy onsets of the surface band (when plotted as a function of k_y along the Γ -Y direction).”

3. Related to the previous question, does the band structure experimentally obtained from the standing waves in Fig. 2e really show split bands? It can hardly be seen in the figure because the calculated bands (blue curves) are superimposed.

We agree with the referee that it was difficult to distinguish those details and, therefore, we included the Fourier transformed dI/dV data without superposed curves in SI as **Fig. S14a** (please see also answer to point 5). On this plot it is easier to identify in the obtained band the higher intensities at k_y^{\min} and k_y^{\max} starting from ~ 0.3 eV, which follows the higher density of states at the bottom and top of the CBE surface band when the different contributions due to the dispersion along the direction perpendicular to the Ge-dimer wires are integrated. We have also modified the original **Fig. 2e** (now **Fig. 3e**) by presenting the computed k_y -resolved density of states (DOS(E, k_y), equation 3 in section **S9** in the **SI**), which confirms that a higher DOS is expected at the energy onsets of the surface band (when plotted as a function of k_y). In the reproduction of **Fig. 3e** below, we indicate the positions of k_y^{\min} and k_y^{\max} for an arbitrary energy to help the discussion here.

Fig. 3e: Left: computed density of states of a twelve-layer $Ge(001)-c(4 \times 2)$ slab (broadened by $\eta = 0.015$ eV) as a function of energy and k_y (i.e., integrating the contribution for all k_x at each point, see equation 3 in the **SI**). Right: one-dimensional Fourier transform of the data presented in **d** reproducing the unoccupied band structure of the Ge dimer wire. Brighter colors correspond to higher intensities.

Furthermore, following our argumentation in section **S9** in the **SI** we find a correlation between $DOS(E, k_y)$ and the Fourier transform of the experimental $dI/dV(y, V=E-E_F)$, giving a clear justification for the experimental maps to be dominated by contributions coming from two distinct wave vectors that map the DOS enhancements at the onsets of the surface bands. These theoretical expectations are fully confirmed by the experiments (see new **Fig. 3e** and **Fig. S14**).

Taking cross section profiles along the wave vector at a few energies should help to find two bands, if the bands are really split.

The detailed analysis of experimental data is included in the extended section S5 with additional figures S14 and S15 (see answer to point 5). As requested below we present two examples of the corresponding cross-sections.

Fig. R1. (a) Raw FFT data from **Fig. 3e** and **Fig. S14a** with two examples of line profile cross-sections: along k axis (blue line, b) and along energy axis (red line, c). Black arrows on both profiles point two registered bands.

4. Line190: “The dispersion of these bands agrees also with previously reported single-probe STS studies [33, 34, 43] and with a very recent angle resolved two-photon photoelectron spectroscopy experiment [36].” Again, neither of cited papers showed a split π^* band.

We hope that this point has been sufficiently clarified in our responses to points 2 and 3 above. As pointed out by the referee there is no splitting and we have removed this incorrect wording from our manuscript to avoid further confusion. The apparent splitting in our band structures comes from the doubling of the unit cell and plotting the bands along the Γ -Y direction. As explained in our response to point 3, this was justified since this particular cut of the band structure is the most relevant to be compared with the Fourier transform of our STS data mapping the standing waves coming from the Ge(001) surface bands. However, our new

version avoids this potentially misleading comparison and uses the more relevant k_y -resolved DOS.

5. To plot band dispersions in Fig. 2e from standing waves, the authors used cosine waves with two wave vectors (SI Section S5). However, three k vectors are plotted at 0.65 eV and 0.7 eV. How did they do this?

We thank the reviewer for pointing this inconsistency. As noticed by the reviewer our model could capture only two frequencies. Including a third cosine term would introduce additional three parameters, which would make our model overparameterized. Thus, we do not present three k values for any energy. Please note that for energies between 0.65eV and 0.7eV the captured k values could come either from band #1 or #3, never from both at the same energy. This is clarified in the extended S6 section of SI, which includes the additional figure **S14**. This figure presents data from new **Fig. 3e** in more detail. **Fig. S14** contains also additional points taken from dI/dV maps cross-sections presented in the new **Fig. 3b**.

The modified last part of section 6 of SI now reads (previous lines 273-294):

*“The complete reflection of quasi-particles for the energy range of interest allows to use the back-scattering process to experimentally reconstruct the band structure of the Ge dimer wires [2,18,19]. Reconstruction of Ge(001)-c(4x2) band structure along Ge dimer rows is done by introducing a scattering vector q with the length of the double length of the corresponding quasi-particle wave function vector k according to the following formula $|q| = |k_i - k_r| = 2k$, where k_i and k_r are the incident and reflected quasi-particles wave vectors, respectively. The experimental data in **Fig. 3d** from the main text comprise spectroscopy points acquired for 512 energy (voltage bias) points from 0 to 0.9 eV at 47 positions equally spaced by 0.4 nm on the same Ge dimer row. The one-dimensional Fourier transform (power spectrum) of these data presented in **Fig. 3e** as a background (right part) was done with the use of WSXM software [20]. In a raw form it is shown in **Fig. S14a**. The points describing the dispersion of the surface bands in **Fig. 3e** and **Fig. S14b** were obtained by a separate analysis of the spatial pattern for each energy value. First, to reduce noise, the signal from ten adjacent energy points were averaged out. This led to the energy resolution of about 0.018 eV. Then, for every line we fitted a trigonometric function multiplied by a dying-off exponential function: $f(x) = A \exp(-x/L)(\cos(2k_1x - \phi_1) + B \cos(2k_2x - \phi_2)) + C$, where A , B , C , L , ϕ_1 , ϕ_2 , k_1 and k_2 are parameters. In this formula we generally assume two dominant wave vectors. We assume here that the interference patterns are independently formed for each band. For energies below +0.35 eV we used only single cosine function ($B=0$) following Nakatsuji et al. [2], Sagisaka and Fujita [18]. Similar procedure could be performed for cross-sections of constant current dI/dV maps presented in **Fig. 3b**. In this case the spatial resolution of data is increased to 0.07nm but the additional term related to unit cell modulation must be also included. It has $D \cos(k_3x - \phi_3)$ form, where k_3 is in the order of 8/nm (~ 0.8 nm modulation). $E(k)$ points resulting from these fits are presented in **Fig. S14c**. **Fig.***

S15 presents the examples of raw dI/dV data for chosen constant energy cross-sections (a-c) and dI/dV constant current maps cross-sections (d-f) with corresponding fits.

New **Fig. S14**:

Fig. S14. One-dimensional Fourier transform (FT) of the data presented in **Fig. 3d**. (a) Raw FT data. (b) FT data with superimposed square, triangular and circle $E(k)$ points obtained by fitting procedure applied to constant energy cross-sections of dI/dV data from **Fig. 3d** (see examples in **Fig. S15a-c**). Note that our procedure could only capture two wave vectors. (c) FT data with superimposed $E(k)$ points obtained by fitting procedure applied to constant current dI/dV maps from **Fig. 3b** (see examples in **Fig. S15d-f**). Error bars correspond to statistics obtained on 25 dI/dV line cross-sections. Note good matching between both type of point data and FT background.

I would like to see dI/dV cross section profiles along the dimer row and their fitting curves.

We present the corresponding fitting curves in new figure **S15**.

New **Fig. S15**:

Fig. S15. (a-c) Examples of dI/dV constant energy cross-sections from data presented in **Fig. 3d** (green curves) with corresponding fits (red curves). (d-f) Examples of dI/dV constant current map cross-sections from data presented in **Fig. 3b** (green curves) with corresponding fits (black and red curves). For lower energies the fits are performed for longer distances (24nm, black curve in **d**) than for higher energies (15nm, red curves in **e** and **f**). Note that single frequency could not reproduce the data in any of the presented dI/dV distance dependence.

In ref 33 and ref 34, previous fitting to standing waves were done with only one k vector. How do the authors explain this inconsistency?

In Ref. 33 (Ref 50 in the new version of the manuscript) (**Fig. 7**) one can see that the quality of the fit using a single wave vector is clearly worse for Ge(001) than for Si(001). Our own data for Ge(001) clearly indicate that at least two different wave vectors are required for an accurate fitting. Regarding Ref. 34 (new Ref. 48), the cross-sections shown in **Fig. 5** (for the $p(2 \times 2)$ reconstruction in that case) also exhibit a complex behaviour similar to ours, which will require certainly more than one \mathbf{k} vector to be correctly described. Indeed, using just two wave vectors can also be considered as an approximation to the real situation. As we explained in our response to point 3, for a standing wave coming from the CBE surface band at energy E we can expect contributions from a continuum of wave vectors in the range from $\mathbf{k}_y^{\min}(E)$ to $\mathbf{k}_y^{\max}(E)$. However, both the presence of a maxima in the DOS for those threshold wave vectors in connection with our simple theory in section **S9** in the **SI**, and the basic mathematical properties of the Fourier transform indicate that we should be able to reproduce the behaviour assuming contributions only from the two extremal wave vectors $\mathbf{k}_y^{\min}(E)$ and $\mathbf{k}_y^{\max}(E)$. In any case, besides this plausibility arguments and theoretical analysis, the ultimate

justification for using two wave vectors in our fitting comes from the experimental evidence as discussed in detail above.

6. Line 114: “Marked as tip1 in Fig.1” should be “Fig.1d.”

The reference refers to the whole Figure 1, as *tip1* is presented in panels **b-f**. Nevertheless, we have included “*tip1*” and “*tip2*” labels on the former panel **d** (now **Fig. 2a**) to make the discussion more clear.

7. While line 114 reads “the first STM probe is kept in a tunneling regime with a junction resistance larger than 100 GΩ,” line 130 reads “Tip1 (blue circle) was kept in a tunneling condition (I=10 pA, Vsample=-0.5 V). This corresponds to 50 GΩ. Furthermore, line 90 (caption for Fig. 1) reads “Filled-state STM images (I=20 pA, Vsample=-0.5 V) obtained prior to the 2P-STs experiment.” This tunneling parameter corresponds to 25 GΩ. Furthermore, I pointed out in my comment 1 that the tunnel resistance is 1.5 GΩ in Fig. 1e. These inconsistencies are very confusing.

We thank the reviewer for pointing out these inconsistencies, which came from our lack of direct definition of the STM contact resistance (originally, we only referred to our previous work in the SI). The new **SI** section **S2** provides our methodology for contact determination (see our answer to the first comment above). Here, all the contact resistances are determined near the Fermi energy and we now refer to them as “low-bias junction resistances”. In this low voltage range the corresponding I-V curves are linear. Moreover, in this range we can define the contacts for both probes. Note that in the case of our drain probe (*tip2*) after approaching the surface by a few Angstroms and for biases exceeding 50 mV, the resulting tunneling currents would affect the local structure of the surface. Such structural instabilities indicate that this contact must be characterized at very low bias. To visualize the difference between the classical multiprobe experimental methodologies and our study, in this section we also confront our tunneling contact regime with a commonly used Ohmic contact.

Additionally, there was one typo in the text noticed by the reviewer. The Z_0 value for *tip2* was indeed defined by 20 pA and -0.5 V.

The modified sentence (previously line 131) now reads:

“Starting from comparable feedback conditions (20 pA, -0.5 V), tip2 (white circle) was approached down to the surface by 4 Å, resulting in a final ~50 MΩ low-bias junction resistance.”

8. Where were the tip1 and tip2 positioned exactly? If the tips are assumed to be positioned at the center of blue and white circles in Fig.1d, the tips were over the up atoms of the Ge dimers because the image was taken in filled state. But the transconductance measurement was carried out over the pi* state, which is predominantly distributed over the down atoms of the dimers. Is there any specific reason to have chosen the up atoms?

We thank the reviewer for pointing the atomic precision of our two-probe experiments. We always tried to position the tips in between the dimer atoms, thus centrally over the Ge dimer row. To underline this fact, we added two small crosses inside blue and white circles in new **Fig. 2a**.

9. The dI_2/dV_1 spectrum in Fig.1f shows a small peak at 0.1 V, which has almost the same magnitude as the peak at 0.35 V. It also appears in Figs. S1 and S2. Do the authors have any comment on this peak?

We do not know the origin of this peak; however, it is also present in single probe dI/dV data obtained over $c(4 \times 2)$ reconstruction of Ge(001) if tip-to-sample distance is relatively small. See for example **Fig. 2b** from [53] and **Fig. 3** from [47]. We think that its origin may result from processes not related directly to the electronic band structure of bare Ge(001)- $c(4 \times 2)$ surface and thus its analysis is beyond the scope of this work.

We added the corresponding comment to the **Fig. 2** caption:

“The small peak in dI_2/dV_1 spectrum located around 0.1 eV has unknown origin. This peak is also registered by single probe STS experiments performed on Ge(001)- $c(4 \times 2)$ surface [47, 53].”

10. A schematic of the Brillouin zone would help the reader to understand the band structure in Fig.1a.

We have included a scheme for the Brillouin zone as an inset in **Fig. 1a** according to the unit cell used (also shown in the inset).

11. Line 234: “In our self-consistent multi-terminal treatment, we considered a model system composed of a six-layer Ge(001)- $c(4 \times 2)$ slab contacted...” The authors used a 12-layer slab, didn’t they?

We thank the reviewer for calling our attention for this typo, since we should have written “six-bilayers” as we did in the Supporting Information (SI). Although there is no standard way to define the number of layers in Ge/Si, we found that most often it is defined as the reviewer suggested with layers instead of bilayers and, therefore, we have changed it accordingly.

In the manuscript, we have substituted “six-layer” by “twelve-layer” in the caption of **Fig. 1** (former line 80) and at line 234 quoted by the reviewer.

We have also modified the following in the SI for consistency:

- line 115, where it was written “six Ge bilayer” now reads “twelve Ge layer”;
- captions of **Fig. S16** and **S19** where written “six-layer Ge” now reads “twelve-layer Ge”;
- at line 402 where written “six-layer Ge” now reads “twelve-layer Ge”;
- at line 409 where written “4-layer bulk Ge” now reads “8-layer bulk Ge”;
- at line 410 where written “14-layers below” now reads “28-layers below”;

- at line 411 where written “10-layer Ge” now reads “20-layer Ge”;
- at line 418 where written “six-layer slab” now reads “twelve-layer slab”;
- caption of **Fig. S20** where written “six-layer Ge” now reads “twelve-layer Ge”.

12. Line 263: “Furthermore, the wave functions corresponding to the CBE+1 bands present a strong phase modulation between neighboring dimers (Fig. S14). Therefore, an s-symmetry tip wave function in the tunneling limit is expected to couple weakly to this band [44], which will contribute to further reduce the signal from this peak as the tip-to-surface distance is increased.” I cannot agree with this statement. Fig. S13(c) shows that the main orbital producing the CBE+1 band is P_z , which is more likely to interact with the s orbital of the tip than with the other orbitals. How is the coupling of wavefunctions between the tip and the dimer influenced by the phase of the wavefunction of an adjacent dimer?

The statement is based on similar effects observed in STS measurements of graphene nanoribbons (see H. Söde *et al.* Phys. Rev. B **91**, 0458429 (2015) and L. Talirz *et al.* ACS Nano **11**, 1380-1388 (2017)), which are also dominated by p_z contributions. These works show that states whose wave function sign oscillates (i.e., that present many nodal planes perpendicular to the ribbon’s plane) across the ribbon only give rise to a noticeable intensity localized in the ribbon edges in constant current dI/dV maps. However, over the central region the STS maps exhibit a highly reduced intensity with respect to the expectations based on simple Tersoff-Hamann theory. This is due to the interference of different current contributions collected by the tip with different phases. Notice that standard metallic STM tips are characterized by a single prominent atom with spherical symmetry s orbital that probes the evanescent electronic states of the sample several Angstroms away from the surface. Thus, they collect current contributions from different atoms in the surface within regions of tens of Å². For this reason, states with strong sign oscillations are hardly detected in STS measurements at typical tip-ribbon distances. Indeed, the key role of the nodal planes perpendicular to the surface for STM imaging of adsorbed molecular species has long been recognized (see, for example, P. Sautet, Chem. Rev. **97**, 1097 (1997)).

In the Ge(001) surface, the wave function corresponding to the CBE+1 band shows a strong sign oscillation along the dimer wire. Therefore, given its large p_z character (as in the graphene nanoribbons) a fast decay rate of the STS signal is also expected over the dimer row. To further support this observation we performed STM simulations using the SGFM method (J. Cerda *et al.* Phys. Rev. B **56**, 15885 (1997)) with an extended Hückel molecular orbital Hamiltonian parameterized by fitting accurate DFT band structures. This methodology was optimized to simulate STM images on Ge(001), where it provides images in excellent agreement with experiment (M. Kolmer *et al.* Phys. Rev. B **86**, 125307 (2012)). The simulation setup is shown in the new **Fig. S18a**, where the STM tip is modeled as a semi-infinite W(111) slab with a Ge-terminated tip address a Ge(001)-c(4x2) surface defined by nine layers on top of a semi-infinite Ge(001) bulk. The calculated transmission functions for different tip-to-surface distances are presented in **Fig. S18b**, where the peak corresponding to CBE+1 becomes less visible as the tip-to-surface distance is increased. These calculations are in qualitative agreement with our 3-terminal simulations (**Fig. S11c**), where the tip-to-surface

transmission for different distances present a faster decay for CBE+1 as the tip-sample distance increases.

Fig. S18. (a) STM configuration setup used for the SGFM method. (b) Transmission function obtained with the simulation setup in a for different tip-to-surface distances. A peak corresponding to the CBE+1 becomes clearly visible only at short tip-to-surface distances.

The discussion above as well as the figure describing the new calculations were included in SI section S8.

13. Among simulation data in Fig. 3 b-d, there is consistency in specific energies (0.3 eV, 0.7 eV and 0.9 eV) at which characteristic transmission probabilities appear. The authors claim that this is because all these features originate from the opening of new conduction channels such as CBE, CBE+1 and CBE+2 (CBE+2 is rather resonant with bulk, though). Moreover, they found remarkable agreement between the experimental dI_2/dV_1 spectrum and simulation in Fig. 3d, and accordingly the peak at 0.7 V was assigned to CBE+1. On the other hand, the experimental dI_1/dV_1 spectrum features were similarly assigned to CBE, CBE+1 and CBE+2, even though their energies show significant deviation from the simulated energies. I think this is not a reasonable interpretation.

We understand that this is a subtle interpretation and that it involves several observations:

- the calculated CBE and CBE+1 onsets energies agrees well with experiment, however the calculated offset of CBE+2 is underestimated as compared to the experiment (**Fig. 1a** versus the former **Fig. 1f**, now **Fig. 2c**) which can be attributed to limitations in the level of theory used (DFT) as well as finite size effects due to the use of a slab to represent the Ge surface;
- as mentioned by the reviewer, **Fig. 1a** shows that at CBE+2 is resonant with bulk and, therefore, little or no transmission would be expected to the second tip (this process is depicted in the scheme in **Fig. 1c** and is consistent with the experimental observation);
- in **Fig. 3c** (now **Fig. 4c**) we presented the transmission function from tip-to-surface of a 3-terminal setup, with two electrodes defined at the Ge slab terminations and another at the Au tip. Such setup can capture the transmission probabilities from a single tip to

the surface states, however it does not include possible transmission to bulk due to the use of a finite slab (this issue is discussed in detail in **Sec. S10** in the **SI**);

- as discussed in the answer of the reviewer's previous remark, at typical STM tip-to-sample distances, as is the case for *tip1*, the CBE+1 decay faster as compared to the other resonances when pushing the tip away from the surface. Thus, this makes injection of electrons to the CBE+1 resonance from *tip1* less efficient and the experimental identification of the position of CBE+1 in the reported dI_1/dV_1 less clear.

The reviewer's concern have lead us to revise the assignments and specifically the arrows in **Fig. 3c** (now **Fig. 4c**) . Now we also indicate with a dashed arrow the pathway to the bulk Ge states (albeit absent in our slab simulations). Moreover, in the 3rd paragraph of "Transport calculations" section (previous line 260), we have included the sentence:

*"As mentioned above, the CBE+2 is resonant with bulk states which probably dominates the $-dI_1/dV_1$ spectra at the corresponding energies (dashed arrow in **Fig. 4c**), however the bulk states are absent in our 3-terminal setup (see discussion in **SI Sec. S10**)."*

14. Fig. S2: I think this is a very important test to understand the 1D nature of the dimer row and verify the credibility of the transconductance measurement. Can the authors provide a more systematic measurement for different dimer rows such as row0, row-1, row-2, row-3, ...? Why does the magnitudes of the dI_1/dV_1 spectrum vary with the position of the tip1 over different dimer rows?

We thank the reviewer for raising these points, which also correlates with the third comment of Reviewer 3. In the revised version of the manuscript we include more detailed analysis of energy-dependent transport dimensionality in the main text as a new section **Surface states transport dimensionality** containing a new **Fig. 5**:

Fig. 5. Electronic transport along neighboring Ge-dimer rows.

(a) Planar transconductance dI_2/dV_1 STS signals as a function of tip1 voltage obtained for tip1 (source probe) located at different Ge dimer rows with respect to tip2 (drain probe). The corresponding separation across reconstruction rows are indicated on the label (0 marks the same reconstruction row). During acquisition of data the sample was grounded and the tunneling contact resistance of the tip2-sample junction was established to be $\sim 50 \text{ M}\Omega$ and kept constant. Tip1-sample distance was established in all cases at Z_0 defined by $I=20 \text{ pA}$ and $V_{\text{sample}}=-0.5 \text{ V}$. The lateral distance between probes was about 31 nm in each case except reference. (b) Corresponding vertical $-dI_1/dV_1$ STS signals registered simultaneously with the data presented in a. The blue curves presented in a and b show reference data registered by tip1 positioned 13 reconstruction rows apart from tip2. (c) Density of scattering states incoming from tip1 (tip1-resolved DOS) projected on each of the four Ge dimer rows defining the simulation cell (for tip-to-surface distances $D_1=D_2=4.5 \text{ \AA}$). (d) From left-to-right, tip1-resolved LDOS obtained at 0.28 eV (CBE onset), 0.5 eV (quasi-1D region), and 0.72 eV (CBE+1 onset). The radius of the circle centered at each atom is proportional to the calculated lead-resolved DOS. The lateral position of tip1 (tip2) apex atom are marked with

black (red) crosses. In all calculations the Ge slab VBE has been used as a common energy reference.

We have also replaced the original **Fig. S2** with a more detailed new **Fig. S3**.

Here, we would like to stress two main constraints during position dependent two-probe STS experiments performed on Ge(001) surface. As we underline in section **S1** of **SI** the main constraint lies in the atomic scale defects present on the Ge(001) surface, which as intuitively expected influence our coherent transport. Note that generally UHV preparation protocols for Si(001) or Ge(001) surfaces result in defect densities in the limit of our experimental requirements (about 40 nm by 10 nm surface areas without single defects). In fact most of Si or Ge(001) surfaces reported in the literature do not meet this requirement (see for example [Sagisaka K., & Fujita D., *Standing waves on Si(100) and Ge(100) surfaces observed by scanning tunneling microscopy*. Physical Review B, 72(23), 2005]). Here, we put a lot of care to prepare high quality Ge(001) surfaces (**Fig. R2.a**). However, SEM-induced defects inevitable during two-probe experiments decrease the quality of our samples (see **Fig. R2b-d**), which in the end typically exhibit a quality as the surface presented in **Fig. R2b**. Second important aspect lies in the exact geometry of probes. At tip-to-tip distances below 50 nm these geometrical constraints strongly hinders flexible probe positioning. Thus, the acquisition of a reliable set of position dependent data is very challenging.

[redaction]

[redaction]

Fig. R2. Defects on Ge(001) surface. **(a)** STM image (-2V, 100pA) of the Ge(001) surface used in the present study after UHV preparation without any exposure to SEM electron source. **(b-d)** Sequence STM images (+1V, 200pA, $100 \times 100 \text{ nm}^2$) of the Ge(001) after 5 keV e-beam irradiation under SEM. The fluence in **b-c** correspond to irradiation by beam of about 100 pA at 10k magnification (typically used conditions for positioning of STM probes in our setup) for approximately $\Delta t=90\text{s}$ (**b**) and 600s (**c**). Note the same area of STM imaging. The experiments reported in this work are performed on surfaces with quality resembling data shown in panel **b**.

New **Fig. S3**:

Fig. S3. 2P-STs characterization of $c(4 \times 2)$ reconstructed Ge(001). **(a)** Filled-state STM image ($V_{\text{sample}} = -0.5$ V, $I = 20$ pA) obtained prior to the 2P-STs experiment. STM *tip1* positions during 2P-STs are marked by different color squares (white ellipse marks points from **Fig. 5 a,b**). Position of *tip2* is marked by white circle. The increase of probe-to-probe distance during characterization of different relative rows is due to probe geometric constraints. **(b)** Planar transconductance dI_2/dV_1 2P-STs signals as a function of *tip1* voltage obtained for STM probes located at different reconstruction rows (see **a**, blue reference spectrum was obtained outside the presented image). The corresponding rows of separation are indicated on the label (0 is the same Ge dimer row). During acquisition of data the sample was grounded and the tunneling contact resistance of the *tip2*-sample junction was established at ~ 50 M Ω and kept constant. *Tip1*-sample distance was established in all cases at Z_0 defined by $V_{\text{sample}} = -0.5$ V and $I = 20$ pA. We observe strong suppression of transconductance dI_2/dV_1 resonance at 0.35 eV while increasing separation across reconstruction rows between injection and detection of carriers, what confirms its quasi-1D character. The 0.7 eV resonance was in this case still preserved even at 7 rows of separation.

The changes in the dI_1/dV_1 magnitude by moving *tip1* to other Ge-dimer rows while keeping *tip2* fixed is also discussed in the new section **Surface states transport dimensionality**. However, at this stage we cannot provide a conclusive reason for this effect.

15. Fig. S3: The authors claim that the coherence length of electrons in the π^* state is about 50 nm. dI_2/dV_1 measurement should be done with the tip1-tip2 separation over 50 nm to support this and also to see the performance of 2P-STs.

The coherence length value of about 50 nm is extracted from our single-probe measurements near the step-edge. In two-probe experiments we barely register a transconductance signal for such large tip-to-tip distances. However, as discussed above the detailed analysis on larger tip-to-tip distances is strongly affected by the presence of defects. Additionally, we should expect a general reduction of transconductance signal as we increase the *tip1-tip2* distance, not only due to the leakage into the bulk, but due to the non-zero probability to propagate along the direction perpendicular to the Ge-dimer wires (transport is strongly anisotropic but not strictly 1-D on this surface as discussed in detail below in response to the second comment of the reviewer 3).

We include transconductance spectra obtained on distance above 50 nm in the revised version of **Fig. S4c**.

Moreover, the shape of dI_2/dV_1 spectra in Fig. S3c do not look the same as that in Fig. 1f, which makes the credibility of measurement weaker.

We thank the reviewer for raising this point. Despite variations in the exact electronic structure of corresponding tip apexes, the main reason of distinct spectral shape between data presented in **Fig. S4c** and **Fig. 2c** (**Fig. S3c** and **Fig. 1f** in the original version) come from different *tip2*-surface low-bias resistance values ($\sim 500\text{M}\Omega$ and $\sim 50\text{M}\Omega$, respectively). As already discussed in the answer to point 1, this difference changes the relative intensities between CBE and CBE+1 resonances in the corresponding transconductance spectra. For lower resistances (closer *tip2*-sample distance) we observe higher intensity of CBE+1 resonance. These experimental results presented in the new **Fig. S2b** are also in line with our calculations (see **Fig. S11e,f**). The use of slightly different *tip2*-sample distances was related to different STM apex 2 stability in these two separate experiments.

Reviewer #3 (Remarks to the Author):

The authors present unique experimental current transport measurements at a Ge surface on the nanoscale. However, I have some concerns regarding the interpretation of the data, which should be clarified by the authors:

We thank the reviewer for pointing out the uniqueness of our transport measurements as well as for his/her careful points that we address below.

1. Fermi level of the bulk Ge, surface band bending, tip induced band bending:

In Fig. 1c the Fermi level in the Ge bulk is shown to be located at the valence band, which corresponds to degenerate p doping of the sample. On the other hand the authors mention in the methods section that the sample is undoped, which corresponds to the Ge bulk Fermi level located around mid-gap. In my opinion the interpretation of the results will change depending on this.

As discussed above in the answer of point 1 from the second reviewer, the alignment of the Fermi energy in Ge is indeed an important element for our interpretation. Previous observations show that for a bare Ge(001) surface prepared under vacuum conditions, the Fermi level is pinned at the valence band. This fact does not depend on the nominal sample doping level or type. It was confirmed by several studies including single probe STM, mesoscopic multiprobe transport and photoelectron spectroscopy analysis [46-49]. Particularly very recent work by Du and coauthors [49] shows the same Fermi level pinning effect in their photoelectron spectroscopy data of corresponding valence bands for both p- (see Fig. R3 green curve) and n-type (red curve) doped Ge(001) surfaces.

FIG. 6. Ge 3d and VB spectra for clean n - and p -Ge(001)-(2×1) measured with Al K_{α} x rays, and for as-received p -Ge(001) with a thin native oxide measured with hard (~ 6 keV) x rays.

Fig. R3. Photoemission data of valence bands of different Ge(001) surfaces. Image is reprinted from [Du, Y., et al., *Layer-resolved band bending at the n -SrTiO₃(001)/ p -Ge(001) interface*. Physical Review Materials, 2018. 2(9): p. 094602.]

This point was only mentioned at the simulations details part of the **Methods** section. However, and given the questions posed here by the reviewers, we acknowledge the need to emphasize this point in the new version of **Fig. 1**, where the figure caption now includes the sentence:

“Notice that the Fermi energy of Ge(001) is known to be pinned at the top of the Ge bulk valence band [46-49]. Thus, in our scheme the chemical potentials of tip2 and the surface are aligned.”

Further it seems that things like surface band bending and tip induced band bending are not considered in the interpretation of the data.

As pointed above in the case of Ge(001) surface we observe the Fermi level pinning effect resulting from the presence of valence band electronic states on the Fermi energy. These states effectively screen the electric potential differences resulted from different electron work functions between metallic tips and a Ge surface and thus minimize effects typically present on other semiconducting surfaces during STM/STS experiments. This was one of the practical reasons to use the Ge(001)-c(4x2) surface as the model system for our 2P-STs experimental and theoretical analysis.

We underline this important fact in the **Methods** section:

“The valence band electronic states on the Fermi energy effectively screen also the electric potential differences resulted from different electron work functions between metallic tips and a Ge surface and thus minimize effects typically present on other semiconducting surfaces during STM/STS experiments. This was one of the practical reasons to use the Ge(001)-c(4x2) surface as the model system for our 2P-STs experimental and theoretical analysis.”

2. LDOS oscillations at a step edge, coherent transport: The authors present dI/dV oscillations close to a step edge. The authors use these results to assign the current transport as coherent. However, this assignment is not made from the transport measurements itself, but very indirectly via the observation of the oscillations. Due to this, I would strongly recommend to change the title of the paper from Coherent transport... to Current transport... Moreover, (as the authors mention) these results have already been obtained and published by two groups more than ten years ago. Therefore, I would consider these results more for the supplementary information section than for the main text. The assignment of the transport as coherent is based on these well-known result and should not be mentioned in the title as main result of the current paper.

We thank the reviewer for rising this remark. We agree with him/her that the two-probe experiment itself does not probe the coherent nature of the injected/collected electrons. Instead this was proven by the analysis of the scattering in a step edge from electrons injected by a single-probe. Therefore, we have modified the title accordingly to:

“*Electronic transport in planar atomic-scale structures measured by two-probe scanning tunneling spectroscopy*”

Although the measurement of standing waves on Ge surface is not new (as we stated in the text), the proof of the coherent nature of the transmitted electrons is crucial to sustain the interpretation of the two-probe experiment. For this reason, we decided to keep it in the main text. Less importantly, but still valuable, the detailed reconstruction of the Ge(001) surface states bands from the measured standing waves have never been reported before.

3. Fig. S2

I find the data in Fig. S2 much more interesting than the already known LDOS oscillations at a step edge and would propose to include this into the main text.

The inclusion in the main text of the analysis of the electronic transport along neighboring dimer rows was also suggested by the second reviewer, and we thank both reviewers for this interesting suggestion. This data is now included in the new **Fig. 5** together with a new section **Surface states transport dimensionality** where the measurements are described and discussed in detail. Please, see also the answer for point **14** from the second reviewer.

However, these data should be discussed in more depth: Why is the dI_1/dV_1 decreasing from the black trace to the blue trace?

We thank the reviewer for these questions as they motivated us to deeply discuss former **Fig. S2** in the main text of the manuscript. As shown in the new section **Surface states transport dimensionality** once we move *tip1* only by one reconstruction row from the row occupied by *tip2* we observe a decrease of the injected I_1 current in the empty state energy range (using same tip-sample distance as defined by the same filled-state STM parameters) in the range of ~30%. Interestingly the corresponding $-dI_1/dV_1$ spectra have the same shape (positions of resonances), but due to decrease of I_1 current the corresponding $-dI/dV$ resonances have lower intensities. Moreover, the drop of I_1 current values is not observed directly in the nominal values of I_2 but in the relative proportions between I_2 current values (see dI_2/dV_1 resonance trend). In other words, I_2 current values also drops by about 40% for CBE and about 20% for CBE+1 but in the nominal values I_2 are still in the range of 10% of I_1 (see **Fig.R4**). Thus, presence of *tip2* enhances the total I_1 current if both tips are located on the same reconstruction row.

The ultimate reason for this is not clear to us at this stage. One plausible mechanism is that varying the position of the biased *tip1* relative to that of the grounded *tip2*, changes the electrostatic potential distribution in our nanoscale system (note symmetry for +1 and -1 row, which exclude macroscopic geometry of tip as the reason). That may affect the exact set point Z_0 for the same (I-V) parameters determined for occupied states. As a result, *tip1* position is closer to the surface, when *tip2* are closer and, particularly, located on the same reconstruction row. That could then lead to increase of the I_1 current on the unoccupied side of the I-V characteristics. This argumentation can be supported by similarities between these data and results presented in **Fig. S2**, where *tip1* was placed closer to the surface by changing the

current setpoint used for Z_0 determination (while $tip2$ was kept fixed). On the other hand, it is very important to emphasize that it is a highly non-trivial question since there are not obvious effects of electrostatics in other measured quantities. Namely, energies of peaks in the dI/dV are unaffected and even more importantly, as pointed in **S4 section**, capacitance effects have only minor effects as the behavior is seen also in DC current signals (see **Fig. R4**). Finally, the inelastic processes may also contribute to this effect. Please see our general answer to the last remark and the last two sentences of the **Surface states transport dimensionality** section, which read:

“Additionally, even if the transport would be mostly coherent, we should always expect increased signal in the dI_2/dV_1 at the band minima (CBE and CBE+1 resonances), since they get populated by the fraction of electrons inelastically scattered within each surface band [46]. This easily explains the lack of direct proportionality between $-dI_1/dV_1$ and dI_2/dV_1 .”

Thus, as the experimental and theoretical approaches presented in this work are completely novel, we need to acknowledge that detailed explanation of all effects observed in our 2P-STs spectra is beyond our current capabilities.

Because of these limitations in the new section **Surface states transport dimensionality** we only rationalize on the observation of CBE and CBE+1 resonances in the transconductance spectra registered by tips located on separate Ge dimer rows, which we correlate with the large 2-D character of the band structure close to the band minima (CBE and CBE+1 resonances).

Fig. R4. Comparison between currents registered on both probes during 2P-STs experiment performed on consecutive Ge dimer rows. The corresponding dI/dV data are presented in **Fig. 5.a,b**.

Should the dI_2/dV_1 data be scaled by the peak height of the dI_1/dV_1 data as reference? In this case the red trace would become as high as the black one.

We thank reviewer about this suggestion. However, regarding the complexity of experimental data and in line with the above explanation of corresponding signals origin, at this stage we do not see direct reason of scaling dI_2/dV_1 by $-dI_1/dV_1$ data. These signals are measured during the same experiment, but both carry different information about the electronic structure of our system. Their division would result in even more complex quantity.

Alternatively, one may consider a different 2P-STs experimental design, where *tip1* Z_0 would be adjusted to keep the injection current I_1 constant for all used voltages and lateral positions (constant current data). That would require involving feedback loop to position our source probe during acquisition of spectroscopic data. However, the possible interference between feedback loop and AC bias applied for lock-in amplifier signal detection complicates in this case practical realization of this concept.

Why is the CBE +1 not visible in the dI_1/dV_1 spectrum (This is also the case in Fig 1f)?

At the large tip-sample distances that characterize the *tip1-surface* tunneling junction, the dI_1/dV_1 spectra the CBE+1 resonance appears as an elbow in the CBE+2 resonance peak. As explained in detailed in our response to point 12 raised by the Reviewer 2, when tip to surface distance decreases the intensity of the CBE+1 peak increases in the dI/dV , which explains some of the differences between the $-dI_1/dV_1$ and dI_2/dV_1 curves.

The data may be also interpreted in a much simpler way as an anisotropic conductivity of the surface states with higher conductivity along the dimer rows and lower conductivity perpendicular to the dimer rows.

Indeed, we interpret our data as anisotropic transport along the Ge dimer reconstruction rows, what was probably not stressed in the original version of the manuscript. We pointed this fact explicitly in the last paragraph of the section **Surface states transport dimensionality**, which reads:

“The experimental data from Fig. 5a,b could be thus interpreted in the following way. 2P-STs data captures CBE and CBE+1 resonances, which are related to opening of two quasi-1D transport channels along single Ge dimer row. Interestingly, at the energies of the CBE and CBE+1 resonances (band onsets) the corresponding bands have a non-negligible 2D character, thus even though the transport is highly anisotropic, the corresponding resonances are expected to be registered also on the neighboring rows of reconstruction.”

As mentioned in the paper and several times throughout our response to the reviewers' comments, another important ingredient to interpret our transconductance curves is that the CBE+2 resonance is resonant with Ge bulk conduction band. Thus, electrons originally injected from *tip1* at this energy (or above) are expected to eventually leak into the Ge bulk and, thus, will not be collected by *tip2*. In summary, 2P-STs spectra on Ge(001) are the result of a plethora of different effects: highly anisotropic coherent transport, tip-surface contact transmission, elastic scattering to the bulk possible for energies above ~ 0.9 eV, intraband inelastic decay within the surface bands etc. All of them, combined together, explain the shape of the dI_2/dV_1 curves and why they present clear differences with respect to the dI_1/dV_1 . Thus, 2P-STs has the potential to reveal information about many of those interesting aspects.

In my opinion, the manuscript might be suitable for publication in Nature Comm. if the authors can dispel my concerns regarding the interpretation of the data.

We thank the reviewer for his/her positive and constructive opinion. We hope that our answers above have clarified the concerns and that the reviewer therefore finds our manuscript suitable for Nature Communications.

Reviewers' comments:

Reviewer #1 (Remarks to the Author):

I have reviewed the revised version of the manuscript. The authors appear to have gone carefully throughout all the comments and suggestions raised by the reviewers. The answers provided by the authors look convincing to me. In doing so, during the review process the manuscript has been further improved and clarified. Because of this, I confirm my recommendation for accepting the paper for publication.

Reviewer #2 (Remarks to the Author):

I have read the revised manuscript and the responses of the authors to the comments of all reviewers. The authors adequately answered to most of my questions. However, the revised version of the manuscript with the new section on "Surface state transport dimensionality" and Fig. 5 makes me think that the standing wave part is redundant. The standing waves actually do not provide much information for interpreting any of the 2-probe STS data or reinforcing the discussion for Fig 5. As the manuscript has become more concentrated upon the 2-probe STS technique and its supporting calculations by adding the new section, the standing wave part interrupts the fluency of the discussion. The anisotropic character of the π^* state along the dimer row is found in the band diagram in Fig. 1a and the literature cited by the authors. Moreover, although the authors claimed in the responses to the reviewer 3 that "the proof of the coherent nature of the transmitted electrons is crucial to sustain the interpretation of the two-probe experiment. For this reason, we decided to keep it in the main text," I think dI_2/dV_1 spectra as a function of tip1-tip2 separation in Fig. S4c is much more important to see how far the single dimer row can transmit its electronic information and is the data that the reader will want to see in this manuscript rather than generic standing wave data. Therefore, I suggest that the authors remove the standing wave part from the main text and add Fig. S4c in the main text to improve consistency in discussion and keep the novelty of the manuscript suitable to Nature Communications, as suggested by the reviewer 3.

Another reason that I suggest to remove the standing wave part is that the origin of scattering waves with two different wavelengths is not fully explained. The authors stated "The energy positions and dispersions are in good agreement with the band structure calculations" in line 208, but I do not agree with this statement. To me, the dispersion obtained from STS is almost twice as wide as the integrated dispersion. This indicates that two scattering k vectors do not compare k_{\max} and k_{\min} well. The two values k_{\max} and k_{\min} , that the authors have labelled, are actually derived from the dispersions along the Γ -R' section and J-R + R'-Y sections of the Brillouin zone of the primitive $c(4 \times 2)$ cell (See Figure B). I guess that scattering waves with two different wavelengths come from these high symmetry lines, but a deep discussion is missing, such as the reason that scattering is intense at the γ point and at the edges of the BZ.

The authors corrected their interpretation of doubled dispersions in Fig. 1a to the width of dispersion or k_{\max} and k_{\min} . As mentioned above, the doubled dispersion curves were resulted by band folding. I think it is not safe to use the labels k_{\max} and k_{\min} by calculating only at two k_x points. (By the way, how many k_x points were integrated to obtain the left dispersion in Fig. 3e?) Furthermore, in general, a band diagram should be drawn using an irreducible cell. Otherwise, those which are calculated with a larger cell are apt to cause unnecessary complex band folding and misunderstanding of the character of corresponding electronic structure by the readers, unless the band folding is carefully explained, as the authors misinterpreted in the original manuscript. To explain the 2-probe STS data, the band diagram with folded BZ is absolutely unnecessary. Therefore, I recommend to redraw the band diagram in the BZ for the primitive $c(4 \times 2)$ cell. I still do not agree with the discussion in line 286: "Furthermore, the wave functions corresponding to the CBE+1 bands present a strong phase modulation between neighboring dimers (Fig. S17). Therefore, an s -symmetry tip wave function in the tunneling limit is expected to couple weakly to this band [60], which will contribute to further reduce the signal from this peak as the tip-to-surface distance is increased." As the authors stated in their response, if a metallic tip with an s -orbital collects electrons from neighboring dimers as well, the opposite phase effect could possibly reduce the conductance at the energy of the CBE+1 state. And this effect would be enhanced with

increasing the tip-surface distance. I agree with this. However, the same process should happen when the tip moves to next dimer or any place in the flat surface. As a result, the dI/dV image at the same energy should lose any pattern or image contrast along the dimer rows. Moreover, if the same idea were applied to the bottom of the CBE where two adjacent dimers have the same phase of wave function, the tunneling conductance would increase everywhere, resulting in no image contrast in the corresponding dI/dV image along the dimer rows. However, this is not the case. Fig.3b clearly displays dI/dV images resolving each dimer with in the dimer rows. Therefore, I think that the tip wave function is strongly localized over the atom underneath the tip, which is the origin of high spatial resolution in STM. I cannot judge how accurately the SGFM method in Fig. S18 describes the experiment, but this data makes me curious whether the CBE+1 state becomes prominent in a single-probe STS as the tip is placed closer to the surface.

Did the authors include the procedure to calculate lead-resolved DOS in Fig. 5?

As described above, the manuscript still has minor problems but they do not deny the novelty of the 2-probe STS technique. I think the manuscript is appropriate for publication in Nature Communications if the authors deal with the issues listed the above.

Reviewer #3 (Remarks to the Author):

After going through the response letter, I do recommend publication of the manuscript.

Reviewer #1 (Remarks to the Author):

I have reviewed the revised version of the manuscript. The authors appear to have gone carefully throughout all the comments and suggestions raised by the reviewers. The answers provided by the authors look convincing to me.

In doing so, during the review process the manuscript has been further improved and clarified.

Because of this, I confirm my recommendation for accepting the paper for publication.

We thank the reviewer for his/her recommendation for publication of our work.

Reviewer #2 (Remarks to the Author):

I have read the revised manuscript and the responses of the authors to the comments of all reviewers. The authors adequately answered to most of my questions.

We thank the reviewer for his/her positive opinion on our revised manuscript.

However, the revised version of the manuscript with the new section on “Surface state transport dimensionality” and Fig. 5 makes me think that the standing wave part is redundant. The standing waves actually do not provide much information for interpreting any of the 2-probe STS data or reinforcing the discussion for Fig 5. As the manuscript has become more concentrated upon the 2-probe STS technique and its supporting calculations by adding the new section, the standing wave part interrupts the fluency of the discussion. The anisotropic character of the π^* state along the dimer row is found in the band diagram in Fig. 1a and the literature cited by the authors. Moreover, although the authors claimed in the responses to the reviewer 3 that “the proof of the coherent nature of the transmitted electrons is crucial to sustain the interpretation of the two-probe experiment. For this reason, we decided to keep it in the main text,” I think dI_2/dV_1 spectra as a function of tip1-tip2 separation in Fig. S4c is much more important to see how far the single dimer row can transmit its electronic information and is the data that the reader will want to see in this manuscript rather than generic standing wave data. Therefore, I suggest that the authors remove the standing wave part from the main text and add Fig. S4c in the main text to improve consistency in discussion and keep the novelty of the manuscript suitable to Nature Communications, as suggested by the reviewer 3.

We thank the reviewer for his/her opinion about the structure of the revised manuscript. However, at this stage we would like to keep the current reasoning order in our manuscript, which was approved by all our co-authors and two other reviewers including reviewer 3, who initially raised the point about the manuscript structure. We believe that the entire material included in our manuscript and the supporting information give the detailed and precise description of our research results to the interested reader. The division of this material into main and supporting parts is a secondary issue, which is always done by a subjective decision made by the authors. Please note that it is difficult to satisfy all the possible arguments at this

stage. Here we sustain our former argumentation related to the coherence of charge carriers in our system. Moreover, we would like to underline also that our FT-STs data obtained from the step-edge reflection experiment, which are included in the section 2 show that the electronic structure of the chosen model system - Ge(001)-(4×2) surface states - could be described quantitatively by our combined experimental and theoretical approach. This section gives strong fundamentals to support our main conclusions included in the following parts of the manuscript. Finally, we stress again that, although these results present known and documented phenomenon, its detailed characterization by single probe STS spectra at LHe temperature on a high quality Ge(001)-(4×2) surface had not been reported previously.

Another reason that I suggest to remove the standing wave part is that the origin of scattering waves with two different wavelengths is not fully explained. The authors stated “The energy positions and dispersions are in good agreement with the band structure calculations” in line 208, but I do not agree with this statement. To me, the dispersion obtained from STS is almost twice as wide as the integrated dispersion. This indicates that two scattering k vectors do not compare k_{max} and k_{min} well.

In **Fig. R1** we re-plot the data presented in **Fig. 3e** to demonstrate that there is a good agreement between the experimental and computed widths of the surface bands. As described in section **S9**, the k-resolved DOS is calculated as a function of energy with a Lorentzian function (**Eq. 1** from the **SI**) where a broadening parameter η has been arbitrarily chosen to be 0.015 eV. This parameter is related to the state lifetime and experimentally it is typically influenced by a number of effects such as inelastic processes. Therefore, in order to compare the calculated band width with the experiment one should consider the distance between the two intensity maxima appearing at the bottom and at the top of the band, rather than the overall apparent band width. One possible source of confusion here can be the small misalignment of the bottom of CBE between the calculation and experiment. In the figure below we first show the experimental data (same as **Fig. 3e** in the main text) in the left-top panel and in the right-top panel we superimposed the values for the wave vectors obtained from fitting the interference patterns using two Fourier components. These experimentally fitted bands agree well with the position of the intensity maxima in the STS experimental maps. The green bar provides an estimate of the experimental band width (distance between maxima) for a given value of k_y . In the left-down panel the very same green bar is now plotted together with the theoretical k-resolved DOS at the same k_y . The blue curves here correspond to the computed band along the $\Gamma - Y$ direction of our orthogonal supercell (**Fig. 1a** main text), and correspond closely to the position of the maxima of the theoretical k-resolved DOS. Finally, the right-down panel shows the experimental data together with the theoretical bands. This analysis clearly shows that the overall agreement between the experimental and theoretical surface band dispersions is rather good, at least as far as it can be expected using density functional theory within GGA, and sufficient to interpret our experimental observations.

Fig. R1. Top panels: Fourier transform of the experimental STS data close to a step-edge (same as in **Fig. 3e** in the main text). The green bar highlights the surface-band width at a particular k_y . Bottom panels: (Left) comparison of theoretical results for DOS(E, k_y) and the experimental band width (green bar). (Right) Experimental data together with the calculated $k_y^{\min}(E)$ and $k_y^{\max}(E)$ curves (blue curves). This comparison highlights that, besides a slight shift to larger energies of the theoretical results, the band width and dispersion of the surface band are in quite good agreement to the experiment.

The two values k_{\max} and k_{\min} , that the authors have labelled, are actually derived from the dispersions along the Γ -R' section and J-R + R'-Y sections of the Brillouin zone of the primitive $c(4 \times 2)$ cell (See Figure B). I guess that scattering waves with two different wavelengths come from these high symmetry lines, but a deep discussion is missing, such as the reason that scattering is intense at the gamma point and at the edges of the BZ.

Regarding the referee's first comment, a new Figure in the Supporting Information (**Fig. S16**) describes in detail the relation between the band structure computed using the primitive hexagonal cell and the double-sized orthogonal cell used in our work. We find the latter more convenient to interpret the experimental information (one lattice vector perpendicular and the other parallel to the dimer-row direction).

The referee is right in pointing to the fact that the STS maps coming from the backscattering at the step-edge do not contain only two Fourier components. However, as shown by our simple derivation in Section **S9** of the Supporting Information, the Fourier transform of the

$dI/dV(E,y)$ maps is proportional to $DOS(E,k_y)$ as defined by **Eq. 3** in the SI. This quantity is plotted in **Fig. R2** below and for each energy it peaks at two values of k_y , corresponding to the thresholds of the surface band at each energy [$k_y^{\min}(E)$ and $k_y^{\max}(E)$].

Fig. R2. Density of states integrated over k_x for a twelve-layer Ge(001)-c(4x2) slab (**Eq. 1** and **3** from the Supporting Information, with a broadening $\eta = 0.0015$ eV), calculated as a function of k_y and at $E = 0.42, 0.54$ and 0.66 eV (left to right).

Alternatively, **Fig. S20** (previously **Fig. S19**) also presents information that supports the expectation of two main wave vectors contributing to the interference maps. At fixed energy values ranging from 0.35 eV until 0.83 eV, the $k_y(k_x)$ relation provided by the CBE surface band is two-valued and very flat close to $k_x = 0$. Thus, it is expected to lead to van Hove-like singularities in the k -resolved $DOS(E,k_y)$ at two different k_y 's (consistent with **Fig. R2**).

The authors corrected their interpretation of doubled dispersions in Fig. 1a to the width of dispersion or k_{\max} and k_{\min} . As mentioned above, the doubled dispersion curves were resulted by band folding. I think it is not safe to use the labels k_{\max} and k_{\min} by calculating only at two k_x points. (By the way, how many k_x points were integrated to obtain the left dispersion in Fig. 3e?)

We do not base our arguments on the evaluation of only two k_x points, but rather we have used 128×128 equidistant k -point mesh for the entire Brillouin zone shown in the inset of **Fig. 1a**, which was used for the calculation of the DOS shown in the left side of **Fig. 3e** (see also **Eq. 3** and **4** in the Supporting Information). Since our first draft we were concerned with

understanding the complex band-structure of the Ge(001)-c(4x2) and visualizing the dispersions in the 3-dimensional space ($k_x \times k_y \times \text{energy}$). This was done in **Fig. S20** (previously **Fig. S19**) by taking slices of the DOS evaluated in the first quadrant of the Brillouin zone (64x64 k-points) for different energies (from 0.23 eV to 1.01 eV, with respect to the valence band edge). Starting from 0.35 eV, the values of k_y^{min} and k_y^{max} can be unambiguously distinguished for the first surface band and similarly for the second surface band starting from 0.75 eV, until they start to overlap at 0.87 eV. Moreover, the interpretation of the band structure is further supported by the new **Fig. S16** presented and discussed in the next point.

We added the information regarding the k-sampling in section **S9**, where the former line 380 now reads:

“**Fig. S20** shows this k-resolved DOS evaluated in the first quadrant of the Brillouin zone *using 64x64 k-points* at selected energies belonging to the energy interval where the empty surface bands reside (between the red dashed lines from **Fig. S17b**)”

Furthermore, in general, a band diagram should be drawn using an irreducible cell. Otherwise, those which are calculated with a larger cell are apt to cause unnecessary complex band folding and misunderstanding of the character of corresponding electronic structure by the readers, unless the band folding is carefully explained, as the authors misinterpreted in the original manuscript. To explain the 2-probe STS data, the band diagram with folded BZ is absolutely unnecessary. Therefore, I recommend to redraw the band diagram in the BZ for the primitive c(4x2) cell.

We believe the choice of using a supercell defined by orthogonal lattice vectors (parallel and perpendicular to the dimer rows) facilitates the interpretation of both 2-probe and step-edge experiments. Nevertheless, we understand that some readers would prefer to visualize the irreducible cell as suggested by the reviewer and, thus, to further clarify what is the equivalent of the chosen path (**Fig. 1a**) in the primitive cell Brillouin zone we included a following new **Fig. S16** in the **SI**:

FIG. S16. (a) Ge(001)-c(4x2) slab primitive cell (top left), whose primitive vectors form an hexagonal lattice, and a double sized unit cell chosen in this work (top right), defined by orthogonal lattice vectors. The corresponding Brillouin zones are represented below, in gray for the hexagonal lattice and in blue for the orthogonal one. (b) Band structure calculated with the primitive cell along the red and green paths depicted in the Brillouin zone shown in a.

We have included a reference for this figure in the caption of **Fig. 1** in the main text and the paragraph in the **SI** previously at line 337 now reads:

“Before analyzing in more detail the characteristics of the surface states, let’s first clarify the consequence of our choice of a double-sized unit cell in the band structure analysis. **Fig. S16a** shows in the left a primitive cell of the Ge(001) surface, which for the c(4x2) reconstruction is defined by an hexagonal lattice, and in the right the unit cell used in this work with orthogonal lattice vectors, which is a more natural choice to compare with the relevant directions of the performed experiments.

In **Fig. S16b** it is presented the band structure of the Ge(001)-c(4x2) slab calculated with the primitive cell along the two paths depicted in red and green at the Brillouin zone from panel a. One can note that the combination of these two paths (*i.e.*, the folding of the green path into the red) is equivalent to the band structure computed using the orthogonal cell (**Fig. 1a** in the main text and **Fig. S17b** below), whose corresponding Brillouin zone is shown in blue in panel a.

We now proceed with the analysis of the empty surface states of a Ge(001)-c(4x2) slab. **Fig. S17** shows the band structure calculated with the chosen orthogonal cell and the density of states on all Ge atoms and projected (PDOS) on those Ge atoms belonging to the dimer rows at the surface (also the projection on different atomic orbitals is shown in this latter case). One can note that the DOS in the energy window defined by the red dashed lines is almost

completely given by the Ge atoms forming the dimers and that it is dominated by a π character ($4p_z$ orbitals of Ge).”

I still do not agree with the discussion in line 286: “Furthermore, the wave functions corresponding to the CBE+1 bands present a strong phase modulation between neighboring dimers (Fig. S17). Therefore, an s-symmetry tip wave function in the tunneling limit is expected to couple weakly to this band [60], which will contribute to further reduce the signal from this peak as the tip-to-surface distance is increased.” As the authors stated in their response, if a metallic tip with an s-orbital collects electrons from neighboring dimers as well, the opposite phase effect could possibly reduce the conductance at the energy of the CBE+1 state. And this effect would be enhanced with increasing the tip-surface distance. I agree with this. However, the same process should happen when the tip moves to next dimer or any place in the flat surface. As a result, the dI/dV image at the same energy should lose any pattern or image contrast along the dimer rows. Moreover, if the same idea were applied to the bottom of the CBE where two adjacent dimers have the same phase of wave function, the tunneling conductance would increase everywhere, resulting in no image contrast in the corresponding dI/dV image along the dimer rows. However, this is not the case. Fig.3b clearly displays dI/dV images resolving each dimer within the dimer rows. Therefore, I think that the tip wave function is strongly localized over the atom underneath the tip, which is the origin of high spatial resolution in STM. I cannot judge how accurately the SGFM method in Fig. S18 describes the experiment, but this data makes me curious whether the CBE+1 state becomes prominent in a single-probe STS as the tip is placed closer to the surface.

We thank the reviewer for his/her deep interest in our results. As we stated in the manuscript in lines 171-173:

“For example, a new dI_2/dV_1 resonance appears at 0.7 V, *i.e.*, in the energy range of CBE+1 (Fig. 1a), which only appears as an elbow in the single-probe $-dI_1/dV_1$.”

in standard tunneling conditions the CBE+1 feature is registered as an elbow of CBE+2 peak (see also black curve in Fig. R3). As this state is located above 0.6 eV from the Fermi level approaching the tip towards the surface with these bias voltage values results in large current densities, which practically hinder its direct observation in single probe constant height dI/dV STS. Nevertheless, one may use constant current dI/dV STS to enhance visibility of this peak (red curve in Fig. R3). In such experiment starting from high positive sample biases the tip approaches towards the surface after passing through energies related to CBE+2 resonance. This procedure enhances the corresponding CBE+1 resonance (arrow). Importantly in single probe STS the peak related to CBE+1 resonance has its position slightly shifted towards higher energies than the corresponding CBE+1 peak in 2P-STs. Note that this experimental observation is fully consistent with our calculations (see Fig. 4c-d and Fig. S11c-f). Finally, we would like to stress that signatures of CBE+1 are also seen in our step-edge reflection experiment, what strongly supports our interpretation of data and manuscript consistency.

Fig.R3. Observation of CBE+1 resonance in single probe STS experiment performed on Ge(001)-c(4×2) surface. Black and red curves correspond to standard constant height and constant current dI/dV spectra, respectively. For the red curve, at the energy of the CBE+1 resonance (arrow), the tip is located closer to the surface than for the standard dI/dV (black curve). The data were collected sequentially at the same location over Ge dimer row.

Interestingly these experimental results agree with the trend observed in both our tranSIESTA and SGFM calculations, indicating that a lower height of the STM tip enhances the intensity of the CBE+1 peak in the STS.

The referee agrees with us that the position and number of the nodal planes in the wavefunction is likely affect the intensity of the tunneling current flowing through such state, and that this effect will depend on the height of the tip. Thus, it can provide a plausible explanation of the observed dependence for CBE+1. However, the referee also raises some concerns about the role that this should also play on determining the contrast of the STM images and whether this is observed or not in our images. Although we think that the wavefunction modulation should not be expected to wash out completely the contrast of the STM dI/dV images, we concede to the referee that this might not be the only effect contributing to the dimming of the CBE+1 resonance as the tip-to-surface distance is increased. Therefore, we have decided to make our statement in this regard milder, including a new sentence in the corresponding paragraph in the main text:

“For the large tip-to-surface distances that mimic the source probe STM/STS experimental conditions, $T_{st}(E)$ around the CBE+1 resonance energy is relatively low. *The ultimate reason*

for the low CBE+1 peak intensity at these large tip-to-surface separations is not completely clear. However, the wave functions corresponding to the CBE+1 bands present a strong phase modulation between neighboring dimers (**Fig. S18**). Therefore, an *s*-symmetry tip wave function in the tunneling limit is expected to couple weakly to this band [60], which will contribute to further reduce the signal from this peak as the tip-to-surface distance is increased. ...”

In any case, we would like to stress that once established (with the help of calculations and experimental results like those shown in **Fig. R3** above) that the change of relative intensity of the CBE+1 peak in the STS is mostly controlled by the tip height, the ultimate reason why this takes place is not crucial for the interpretation of our 2-tip STS experiments.

Did the authors include the procedure to calculate lead-resolved DOS in Fig. 5?

We thank the reviewer for the careful reading. We have included the following sentence at the end of section **S5**:

“The lead-resolved DOS presented at **Fig. 5c-d** are calculated from the diagonal elements of the lead spectral function [14], which for lead *j* and energy *E* is defined as $A_j(E) = G^r \Gamma_j G^{r\dagger}$, where $G^r(E)$ is the scattering region retarded Green’s function and $\Gamma_j = i(\Sigma_j^r - \Sigma_j^{r\dagger})$ is the coupling matrix, with Σ_j^r the lead *j* retarded self-energy.”

As described above, the manuscript still has minor problems but they do not deny the novelty of the 2-probe STS technique. I think the manuscript is appropriate for publication in Nature Communications if the authors deal with the issues listed the above.

We thank the reviewer for his/her conditional recommendation for publication of our work. We believe that the provided answers and manuscript modifications will finally clear all the concerns and satisfy the reviewer.

Reviewer #3 (Remarks to the Author):

After going through the response letter, I do recommend publication of the manuscript.

We thank the reviewer for his/her recommendation for publication of our work.